# Transcription-coupled genetic instability marks acute lymphoblastic leukemia structural variation hotspots

Merja Heinäniemi[1]*, Tapio Vuorenmaa[1,2†], Susanna Teppo[3†], Minna U Kaikkonen[2†], Maria Bouvy-Liivrand[1], Juha Mehtonen[1], Henri Niskanen[2], Vasilios Zachariadis[4], Saara Laukkanen[3], Thomas Liuksiala[3], Kaisa Teittinen[3], Olli Lohi[3,5]*

[1]School of Medicine, University of Eastern Finland, Kuopio, Finland; [2]A. I. Virtanen Institute for Molecular Sciences, University of Eastern Finland, Kuopio, Finland; [3]School of Medicine, University of Tampere, Tampere, Finland; [4]Department of Molecular Medicine and Surgery, Karolinska Institutet, Stockholm, Sweden; [5]Tampere University Hospital, Tampere, Finland

**Abstract** Progression of malignancy to overt disease requires multiple genetic hits. Activation-induced deaminase (AID) can drive lymphomagenesis by generating off-target DNA breaks at loci that harbor highly active enhancers and display convergent transcription. The first active transcriptional profiles from acute lymphoblastic leukemia (ALL) patients acquired here reveal striking similarity at structural variation (SV) sites. Specific transcriptional features, namely convergent transcription and Pol2 stalling, were detected at breakpoints. The overlap was most prominent at SV with recognition motifs for the recombination activating genes (RAG). We present signal feature analysis to detect vulnerable regions and quantified from human cells how convergent transcription contributes to R-loop generation and RNA polymerase stalling. Wide stalling regions were characterized by high DNAse hypersensitivity and unusually broad H3K4me3 signal. Based on 1382 pre-B-ALL patients, the ETV6-RUNX1 fusion positive patients had over ten-fold elevation in RAG1 while high expression of AID marked pre-B-ALL lacking common cytogenetic changes.

*For correspondence: merja. heinaniemi@uef.fi (MH); olli.lohi@ staff.uta.fi (OL)

†These authors contributed equally to this work

Competing interests: The authors declare that no competing interests exist.

## Introduction

In precursor lymphoblastic leukemia, primary genetic lesions often arise *in utero* (*Wiemels et al., 1999*; *Mori et al., 2002*; *Maia et al., 2003*, *Bateman et al., 2015*), while the onset of overt disease requires additional genetic alterations. Whole-genome sequencing (WGS) of ETV6-RUNX1 (also known as TEL-AML1) positive acute leukemias suggested that the secondary lesions are predominantly caused by off-target activity of the RAG complex (*Papaemmanuil et al., 2014*). In a similar fashion, the expression of the AID complex in more mature B cells is implicated in genomic instability and development of lymphomas (*Meng et al., 2014*; *Qian et al., 2014*; *Robbiani et al. 2015*). To date, WGS in leukemia have been reported from several pre-B-ALL subtypes (*Andersson et al., 2015*; *Holmfeld et al., 2013*; *Paulsson et al., 2015*; *Zhang et al., 2012*), resulting in a comprehensive characterization of the underlying genetic alterations. Therefore, the research focus on leukemia genetics is moving into characterization of the mechanisms by which these lesions occur and the consequences of the resulting clonal heterogeneity.

Antigen receptor genes are assembled from discrete gene segments by RAG-mediated V(D)J recombination at sites of recombination signal sequences (RSS) during early lymphocyte

**eLife digest**   Some of the most common cancers found in children are called precursor leukemias, which may start to develop before birth. Cancerous cells often contain alterations to the genetic information in their DNA. In precursor leukemias, the most common genetic changes involve deleting, adding or rearranging segments of the DNA sequence.

Several researchers have sequenced the entire DNA of childhood leukemia cells, with the result that almost all of the genetic alterations linked to these conditions have been catalogued. These efforts have shown that certain DNA regions are particularly affected by mutations, but no one knows why errors occur so frequently in these regions.

Recent evidence also suggests that transcription – the process of producing useful molecules from a stretch of DNA – can play a role in generating genetic alterations. Heinäniemi et al. have now used a technique called global run-on sequencing to measure the extent of transcription in many different types of leukemia cells. This revealed that in the error-prone DNA regions, two processes – called convergent transcription and transcriptional stalling – interfere with transcription. Both processes temporarily leave the normally double-stranded DNA unzipped as two single strands and free of nucleosomes, which makes DNA more vulnerable to breaking. This would explain how pieces of DNA might be lost, added, or moved to cause the genetic errors that lead to leukemia.

Further investigation revealed that two protein complexes called RAG and AID, which rearrange segments of DNA in immune cells, are likely to cause the errors in the vulnerable DNA regions. Different amounts of RAG and AID were present in different subtypes of leukemia cells, and these amounts also varied with the risk classification of the disease. Further studies are now needed to investigate the exact roles of these protein complexes. This could eventually help scientists devise strategies to protect the DNA of people with leukemia from these errors, which could reduce the risk of the cancer reoccurring.

development (*Gellert 2002*; *Schatz and Swanson, 2011*). Cells incorporate multiple strategies to control the action of the RAG complex to appropriate genomic loci: the expression of *RAG1* and *RAG2* is limited to precursor stages of lymphocytes, the activity of the complex is attenuated during S-phase of cell cycle, and RAG cleavage is directed towards RSS pair containing sequences (*Schatz and Swanson, 2011*). The engagement of RAG2 is further limited by the histone modification H3K4me3, which is typically found at transcription start sites (TSS) (*Matthews et al., 2007*; *Teng et al., 2015*). However, RSS and RSS-like motifs are found only at around 7–40% of breakpoints at SV (genomic imbalance, translocation or inversion) sites (*Andersson et al., 2015*; *Papaemmanuil et al., 2014*). Furthermore, the RSS motifs and H3K4me3 occur frequently in the genome suggesting that additional features, possibly even additional complexes including AID (*Swaminathan et al., 2015*), are relevant for the genetic instability underlying leukemia SV.

In lymphomas, AID off-target effects localize to intragenic super-enhancer (SE) and promoter areas characterized by transcription from both strands, i.e. convergent transcription (convT) (*Meng et al., 2014*). Notably, VH gene segment recombination by RAG at the IgH locus coincides with sense- and antisense transcription (*Bolland et al., 2004*), which could be relevant also at off-target sites. Secondly, stalled polymerases, which are found at exons, R-loops and actively paused at TSS regions (*Jonkers and Lis, 2015*), expose single stranded DNA, recruiting AID via Spt5 binding (*Pavri et al., 2010*). Furthermore, the polymerase complex displaces nucleosomes completely or partially (the H2A/H2B moiety), which in vitro promotes cleavage by RAGs (*Bevington and Boyes, 2013*). Despite these intriguing findings, the relevance of transcription-coupled processes has not been systematically characterized, and the clinical relevance of RAG and AID expression in the different leukemia subtypes remains unclear. RNA polymerases engaged into primary transcription across the genome can be measured using Global-Run-On sequencing (GRO-seq) (*Kaikkonen et al., 2013*). Therefore, this method is ideally suited to distinguish features of transcription at SV sites, including convT and RNA polymerase stalling. To this end, we acquired the first patient profiles of nascent transcriptional activity in leukemic blasts representing seven cytogenetic subgroups and performed integrative analysis of various genome-wide profiles and patient transcriptomes.

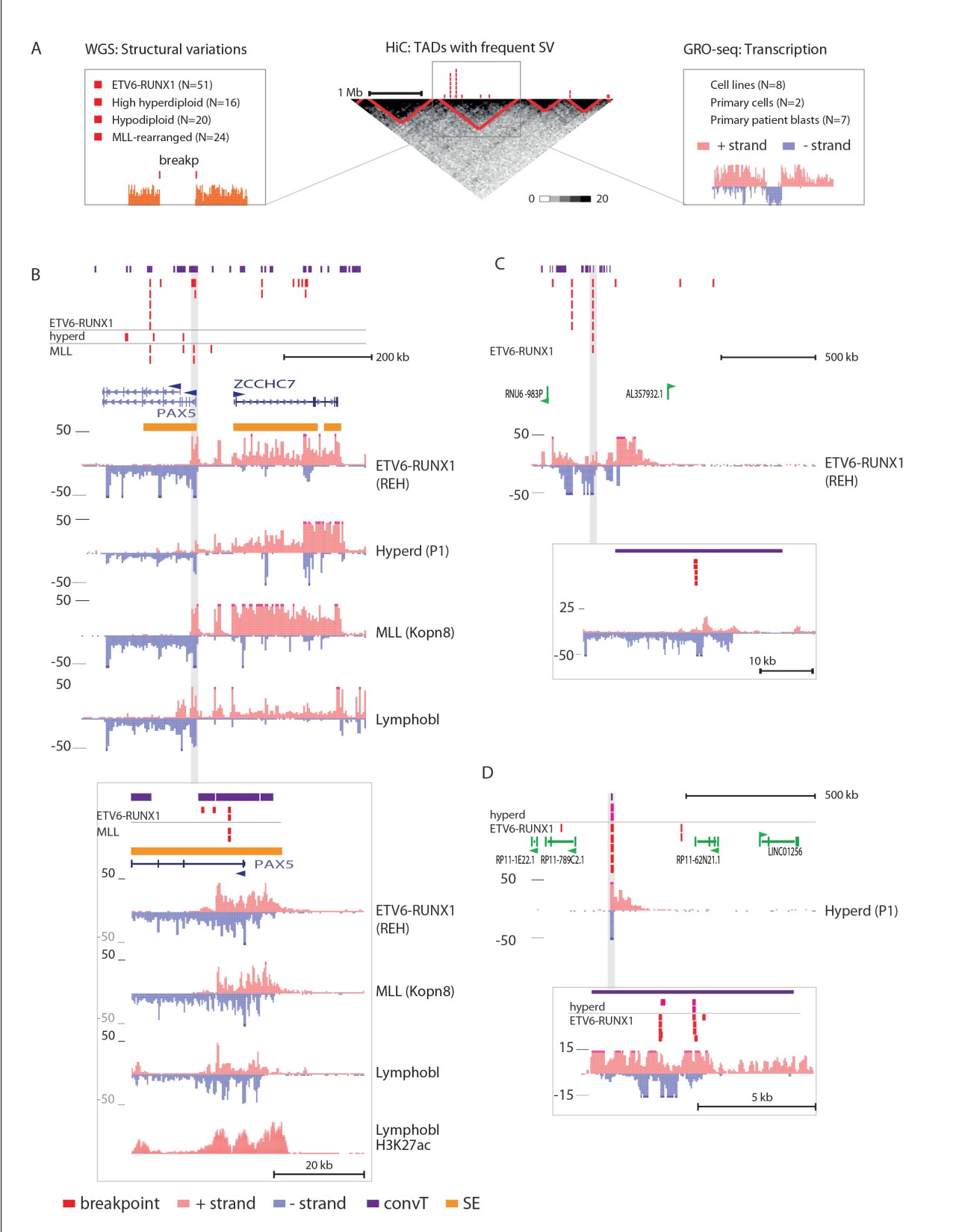

**Figure 1.** Integrative analysis of transcription and high-recurrence SV sites highlights novel transcribed regions. (**A**) WGS data from the ETV6-RUNX1 (51 cases; *Papaemmanuil et al., 2014*), high hyperdiploid (16 cases; *Paulsson et al., 2015*), hypodiploid (20 cases; *Holmfeldt et al., 2013*) and MLL-

*Figure 1 continued*

rearranged (22 cases; *Andersson et al., 2015*) subtypes of precursor B-ALL was integrated with profiles of transcriptional activity assayed using GRO-seq from ALL patient and cell line samples (see also *Figure 1—figure supplement 1* and *Supplementary file 1*). HiC data from B-lymphoid cells (*Rao et al., 2014*) was used to define TADs based on the HiC interaction frequency, shown as grey scale heatmap, in order to distinguish TADs with highest frequency of SV. (B) The *PAX5* and *ZCCHC7* loci are located in the TAD shown that has high SV frequency in hyperdiploid, ETV6-RUNX1- and MLL-fusion positive patients (4, 20 and 6 breakpoints, respectively, *Figure 1—source data 1*). The GRO-seq signal profiles from three pre-B-ALL cytogenetic subtypes and normal B-lymphoblastoid cells are displayed as indicated in the figure (see also *Figure 1—figure supplement 4* and *Figure 2—figure supplement 2*). The y-axis shows the normalized read density (plus strand in red, minus strand in blue). convT regions regions are indicated in purple and leukemia breakpoints in red. The TSS region of *PAX5* overlaps convT that co-localized with an intragenic SE (B-lymphoblastoid H3K27ac track is shown at the bottom). (C) A TAD with the same number of breakpoints (20) in ETV6-RUNX1 patients is shown with signal from REH cells (see also *Figure 1—figure supplement 4*). Genomic annotations include the location of GENCODE transcripts (in green). A strong transcription signal is visible that spans approximately 500 kb near the TAD boundary, lacking annotated transcripts. A zoom-in panel shows the most recurrent SV site. (D) The TAD visualized represents a genomic region that harbors most SV in HeH (see *Figure 1—figure supplement 5* for the hypodiploid SV hotspot). The GRO-seq signal (track from patient 1) indicates a novel locus with abundant transcription in leukemic samples (refer to *Figure 1—figure supplement 4* for all GRO-seq profiles). The highest recurrence of SV occurs at the convT overlapping mid-region (zoom-in panel), which has also two ETV6-RUNX1 breakpoints.

The following source data and figure supplements are available for figure 1:

**Source data 1.** Identified topologically associated domains.

**Figure supplement 1.** Transcriptional activity in leukemic cells from patients, cell lines and primary healthy B-lineage cells is captured in GRO-seq signals.

**Figure supplement 2.** Summary of data used in the integrative analysis.

**Figure supplement 3.** Transcriptional activity in TADs binned by breakpoint frequency.

**Figure supplement 4.** Data from all signal tracks for regions displayed in *Figure 1*.

**Figure supplement 5.** TAD with frequent SV in hypodiploid patients.

## Results

### Integrative analysis of transcription and genomic instability in leukemic cells

Transcriptional activity from ALL cells representing seven different pre-B-ALL cytogenetic subtypes was assayed using GRO-seq (both primary patient and cell line samples, see *Supplementary file 1* and Materials and methods), and jointly analyzed with WGS data from the ETV6-RUNX1 (51 cases; *Papaemmanuil et al., 2014*), high hyperdiploid (HeH, 16 cases; *Paulsson et al., 2015*), hypodiploid (20 cases; *Holmfeldt et al., 2013*) and MLL-rearranged (22 cases at diagnosis and 2 relapses; *Andersson et al., 2015*) subtypes of precursor B-ALL. GRO-seq signals and breakpoint data are shown in *Figure 1—figure supplement 1* at the *CDKN2A* locus, a significant SV site in childhood ALL (*Sulong et al., 2009*).

To systematically identify regions with high frequency of SV across the genome, topologically-associated domains (TADs) were retrieved based on HiC data from B-lymphoid lineage cells (*Rao et al., 2014*). TADs reflect the three-dimensional structure of chromatin. These natural boundaries to transcriptional activity were used to divide the chromosomes into subregions for analysis (see *Figure 1—source data 1* and Materials and methods). To link typical transcriptional activity patterns and hotspots of genomic instability, we related the breakpoint frequency with chromatin domains, as illustrated in *Figure 1A* (see also *Figure 1—figure supplement 2*).

### The most frequent SV regions encompass novel transcribed regions

An increasing trend of transcriptional activity was observed when TADs were compared based on breakpoint frequency quartiles (see Materials and methods, *Figure 1—figure supplement 3*). TADs with highest SV count are shown in *Figure 1* (see also *Figure 1—source data 1* and *Figure 1—*

*figure supplement 4*). The *PAX5* and *ZCCHC7* genes are located within a TAD region with 20 break-points in the ETV6-RUNX1, 4 in HeH and 6 in MLL subtype (excluding the MLL-fusion itself) (*Figure 1B*). Frequent SV were also found in TADs with no annotated coding genes (*Figure 1C*, 20 breakp in ETV6-RUNX1; *Figure 1D*, 4 breakp in HeH), yet GRO-seq exhibited transcription signal spanning several hundred thousand base pairs in both regions, typical of long non-coding transcripts (*Sun et al., 2015*). There was evidence of non-coding transcripts, based on Refseq and GENCODE, but none matched the same location (refer to *Supplementary file 2* for all genomic coordinates shown; a TAD with frequent SV in hypodiploid subtype is shown in *Figure 1—figure supplement 5*). The nascent ALL transcriptomes thus reveal novel transcribed regions as recurrent SV-associated hotspots in the two most common ALL subtypes.

## Convergent transcription and RNA polymerase stalling are prevalent at genomic regions with frequent breakpoint events

The prevailing notion is that active transcription start sites (TSS) in pre-B cells are susceptible to RAG off-targeting due to the H3K4me3 chromatin mark (*Matthews et al., 2007*; *Teng et al., 2015*). How-ever, we noticed that the recurrent breakpoints often lied several kb downstream of TSS, as highlighted in *Figure 1B and D* (see inserts), and coincided with simultaneous transcription on both strands, ie. convT spanning a minimum of 100 bp. In closer examination of the signal data from leu-kemia SV hotspots, many of these regions likely correspond to transcription from intragenic enhancers that generate enhancer RNAs (eRNA) that are typically a few kb in size (*Kaikkonen et al. 2013*). In agreement, a significant enrichment of breakpoints in enhancers overlapping with convT was observed (hypergeometric test P=0.00012 for intergenic and P=4.6e-08 for all enhancers identi-fied based on eRNA signal, see Materials and methods and *Figure 2—source data 1*). An overlap-ping eRNA transcript at the TSS region of *PAX5*, confirmed by the active enhancer chromatin marker H3K27ac, led to convT extending nearly 20 kb, with SV sites located between 3.7–9.7 kb downstream of the TSS (*Figure 1B*, see insert).

Secondly, convT in the vicinity of intragenic breakpoints was often associated with localized eleva-tion in the GRO-seq signal, as exemplified at the *ZCCHC7* and *RAG* loci (*Figure 2A*, see also *Fig-ure 2—figure supplement 1*). The observed signal features were highly reproducible between biological replicates and shared among a subset of cytogenetic groups (*Figure 2—figure supple-ment 2*). We hypothesized that they represent RNA polymerase II (Pol2) stalling events. Previous analyses of Pol2 stalling have focused on promoter proximal regions (*Adelman and Lis, 2012*). To examine such events genome-wide and across gene bodies, we developed a general analysis approach that identifies change points within gene regions and reports those with high elevation in the signal level (see Materials and methods and *Figure 2—source data 1* for the identified regions) (*Killick et al., 2012*). As additional confirmation, we analyzed stalling from Pol2 ChIP-seq in the REH and Nalm6 cell lines (*Figure 2A*). To distinguish between different Pol2 complexes (*Zhou et al., 2012*), antibodies against the serine 2 or serine 5 phosphorylated Pol2 were used (see Materials and methods).

Genome-wide analysis of convT and Pol2 stalling (see Materials and methods and *Figure 2— source data 1*) substantiated the relevance of these observations: considering the breakpoint fre-quency per TAD size, the top ranked TADs in each ALL subtype represented genomic regions with abundant convT and Pol2 stalling (*Figure 2B*). Significant enrichment was confirmed for the upper quartiles (hypergeometric test P=0.00038 in ETV6-RUNX1, P=0.00018 in hyperdiploid, P=0.028 in hypodiploid and P=0.00004 in MLL-rearranged). The increased overlap was found for breakpoints with and without RSS motifs (denoted as R-breakp and NR-breakp, see *Figure 2—figure supple-ment 3* and Materials and methods) and it was preserved when total transcriptional activity was con-sidered (*Figure 2—figure supplement 4*). Furthermore, the distinct transcriptional profile of embryonic stem cells (ES) had lower overlap (*Figure 2—figure supplement 5*).

For comparison, chromatin segmentation of B-lymphoid cells was similarly analyzed (see *Fig-ure 1—source data 1* and *Figure 2—source data 1*). TADs with high number of breakpoints consis-tently had significant overlap with chromatin segments representing active transcription (refer to *Figure 2—source data 1*), supporting a transcription-coupled mechanism for the observed genetic instability. We then distinguished regions with overlap to the transcriptional features defined here within active promoters and enhancers. Comparing these against the TAD SV frequency quartiles

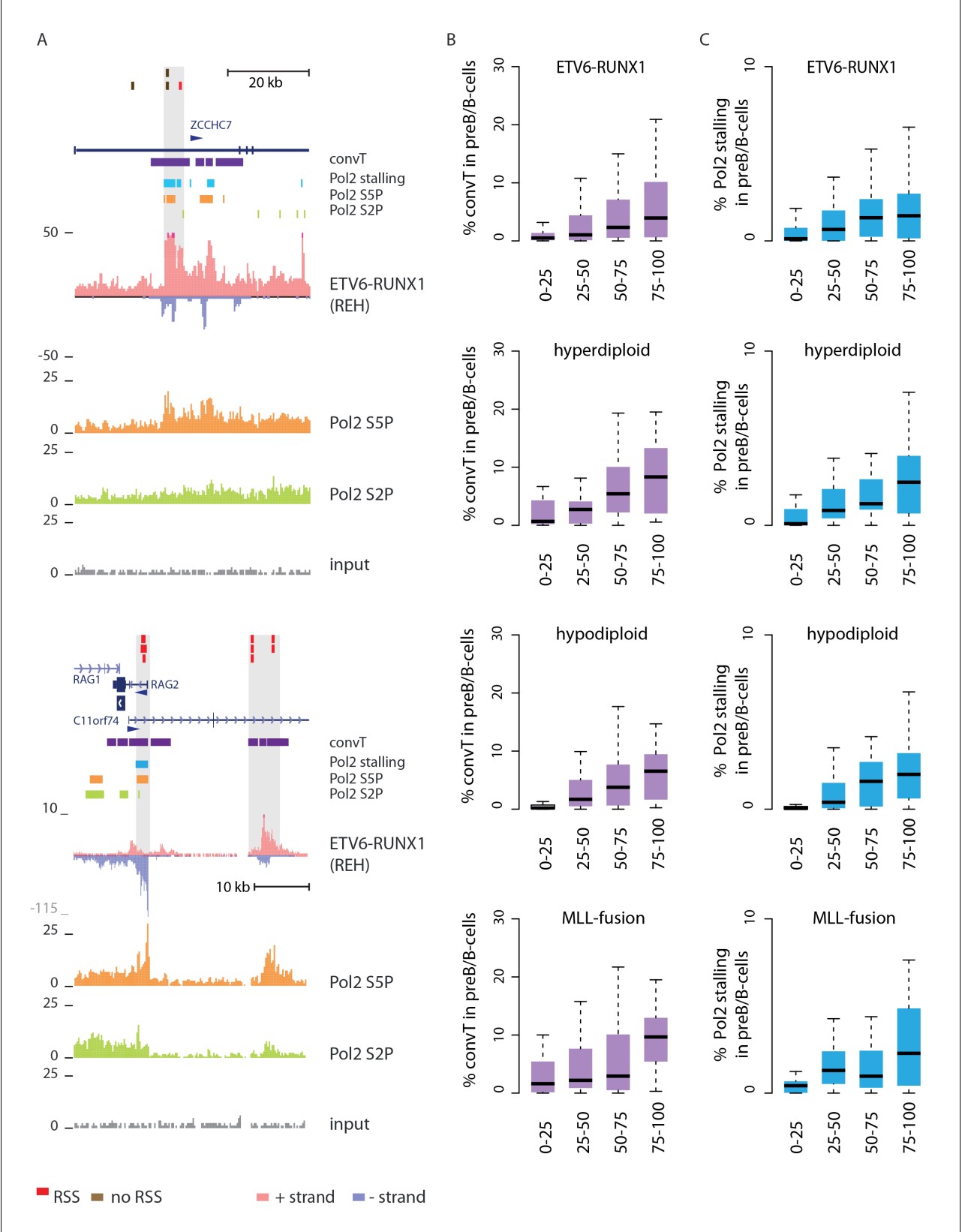

**Figure 2.** Convergent transcription and Pol2 stalling characterize genomic regions with high number of breakpoint events. (**A**) The GRO-seq signal in the ETV6-RUNX1 positive REH cell line is shown to exemplify the co-occurrence of convT (in purple) and local elevation in GRO-seq signal (Pol2 stalling,

*Figure 2 continued on next page*

*Figure 2 continued*

in light blue) at both R- and NR-breakp (in red and brown, respectively) that reside within intronic (*ZCCHC7*), TSS (*RAG2*) or putative enhancer regions (*RAG2*). The elevated signal is also visible in Pol2 ChIP-seq signal (Pol2 S2P in green, Pol2 S5P in orange, input in grey). See also *Figure 2—figure supplement 1*. The percentage of TAD spanned by convT (in B) or Pol2 stalling (in C) in pre-B/B-lymphoid cells is summarized as boxplots from TADs divided into quartiles based on number of breakpoints per bp (see also *Figure 1—figure supplement 3*, *Figure 2—figure supplement 3–6*). The quartile ranges are for exclusive lower and inclusive upper value in the range, as indicated. Refer to *Figure 2—source data 1* for statistical analysis.

The following source data and figure supplements are available for figure 2:

**Source data 1.** Identified convT and Pol2 stalling regions.
**Figure supplement 1.** Data from all signal tracks for regions displayed in *Figure 2*.
**Figure supplement 2.** The GRO-seq signal from replicate samples generated from ALL cells displayed at the *PAX5/ZCCHC7* locus.
**Figure supplement 3.** Signal feature span for TADs ordered separately by R-breakp or NR-breakp frequency.
**Figure supplement 4.** Signal feature span normalized by total transcribed area for TADs sorted by breakpoint frequency.
**Figure supplement 5.** Overlap of TADs with convT in ES cells.
**Figure supplement 6.** TAD analysis using promoter and enhancer chromatin segments stratified by convT and Pol2 stalling.

(*Figure 2—figure supplement 6*), as before, revealed the most pronounced enrichment in convT/Pol2 stall overlapping regions.

Next, we set out to define what may link convT and Pol2 stalling regions with AID and RAG recruitment. The signal feature detection for convT (as in *Meng et al., 2014*) and Pol2 stalling (as defined here) enables this on a genome-wide level.

## R-loop formation and convergent transcription co-occur with Pol2 stalling

RNA polymerases are expected to stall at regions harboring R-loop forming sequences (RLFS) (*Skourti-Stathaki et al., 2014a*; *Jenjaroenpun et al., 2015*). The sensitivity of DNA sequence to form R-loops can be computationally predicted (*Jenjaroenpun et al., 2015*) (see Materials and methods). These RLFS motif containing regions exhibited a significantly higher overlap with Pol2 stalling sites when compared to random intragenic regions (*Figure 3B*, empirical P<0.001 in B-lineage and ES cells). A highly concordant local RLFS motif density and GRO-seq signal profile was observed across gene regions (*Figure 3—figure supplement 1A and B*). The profiles peaked near TSS, where the presence of RLFS motifs led to a significant elevation in the median GRO-seq signal level (*Figure 3—figure supplement 1*, 2.1-fold increase in B-lineage cells, Wilcoxon rank sum test P<2.2e-16, 95% CI 2.1–2.3). As a second mechanism, collisions due to convT may halt transcription (*Prescott and Proudfoot, 2002*) in a dynamic and cell-specific manner. Accordingly, higher antisense signal at convT regions (see Materials and methods) increased the overlap with Pol2 stalling sites on the sense strand (*Figure 3B*), intriguingly exceeding that observed for RLFS motifs (*Figure 3A*).

As an additional experimental validation of R-loops, we used DNA-RNA-immunoprecipitation sequencing (DRIP-seq) results from ES cells (see Materials and methods) that correspond to detection of DNA-RNA hybrids (*Ginno et al., 2013*). The 2.1-fold elevation in median DRIP-seq signal confirmed that RLFS motifs favor DNA-RNA hybrid formation (*Figure 3C*, Wilcoxon rank sum test P<2.2e-16, 95% CI 2.0–2.1, see *Figure 3—source data 2* for each replicate). Moreover, DRIP-seq quantification showed 1.7-fold higher median signal at convT-positive TSS regions (*Figure 3D*, Wilcoxon rank sum test P<2.2e-16, 95% CI 1.6–1.7). These results demonstrate that transcription stalling occurs at RLFS and convT regions in mammalian cells that associates with R-loop formation based on evidence from ES cells.

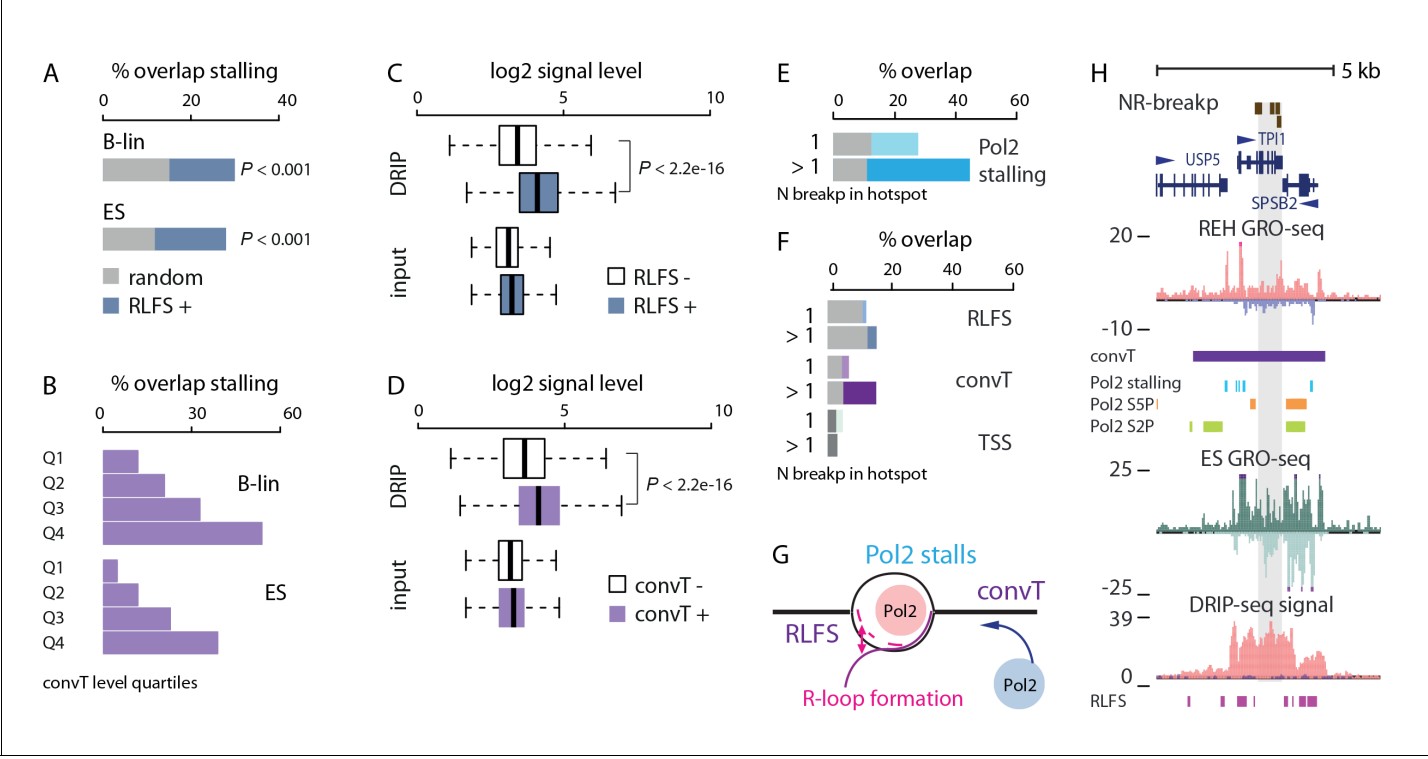

**Figure 3.** Indication of transcription-coupled genetic instability at leukemia SV hotspots lacking RSS motifs. (**A**) Overlap between RLFS motif harboring intragenic regions and detected Pol2 stalling sites in B-lineage and ES cells. The high overlap of RLFS-positive regions is statistically significant compared to random regions (empirical P is indicated for 30% and 28% overlaps, respectively). (**B**) Overlap of detected Pol2 stalling sites also increases based on the strength of antisense signal level for B-lineage and ES cell convT regions divided into quartiles. (**C**) The influence of RLFS at TSS on ES cell DRIP-seq signal level is shown (Wilcoxon rank sum test P is indicated). Input signal levels are shown as control. (**D**) ES cell DRIP-seq signal is plotted similarly as in C, from convT-positive and -negative TSS regions. The DRIP-signal is higher in convT-positive TSS (Wilcoxon rank sum test P is indicated, TSS with convT N = 11774, TSS without convT N = 12092, refer to *Figure 3—source data 2* for statistical analysis based on separate DRIP-seq replicates). (**E**) The percentages of breakpoint regions with no RSS motifs overlapping intragenic Pol2 stalling sites found in B-lineage cells are shown as barplots. The mean overlap observed in random sampling is indicated in grey bars (further statistical analysis is presented in *Supplementary file 3*). Categories with increasing cut-off for recurrence (1: non-recurrent in dim color, >1 and above: recurrent in darker color) were tested. (**F**) Overlap with RLFS, convT and annotated TSS is shown, as in E, for ETV6-RUNX1 NR-breakp (see also *Supplementary file 3*). (**G**) A schematic model illustrating how transcription from both strands (convT) or RLFS can locally arrest the Pol2 complex leading to recruitment of DNA damage-sensing complexes to R-loops, such as AID or BRCA (*Alt et al., 2013*, *Hatchi et al., 2015*), in an RSS-independent manner. (**H**) NR-breakp hotspot with the highest recurrence (*TPI1* locus) is shown. DRIP-seq signal (shown in tones of red overlaid with input control signal in blue), and RLFS motifs indicated as a magenta bar track represent two levels of independent data that were integrated with GRO-seq data (signal from REH and ES cells is shown) to characterize properties of convT and Pol2 stalling regions. The breakpoint data (NR-breakp in brown) and detected convT (in purple) and Pol2 stalling in B-lineage cells (in blue) are shown. At the the recurrent breakpoint sites antisense transcription of neighboring gene (*SPSB2* primary transcript) leads to a broad convT region, as indicated in the figure. Elevated DRIP-signal indicates formation of DNA-RNA hybrids (see also *Figure 3—figure supplement 3*).

The following source data and figure supplements are available for figure 3:

**Source data 1.** Breakpoint clustering to regions.

**Source data 2.** Statistical analysis of separate DRIP-seq and DNAse-seq replicates.

**Figure supplement 1.** GRO-seq, RLFS and DRIP-seq signal profiles across genes.

**Figure supplement 2.** Venn diagrams comparing SV within Pol2 stalling regions based on GRO- and ChIP-seq profiles.

**Figure supplement 3.** Data from all signal tracks for regions displayed in *Figure 3*.

## Transcriptional-coupled instability at RSS-independent SV hotspots

A mechanistic link between R-loops and AID off-targeting has been established in lymphomas (*Alt et al., 2013*). With this in mind, we investigated regions where off-targeting could occur via R--loops by focusing on breakpoints without RSS-motifs (data shown in figures represents the 416 ETV6-RUNX1 NR-breakp, refer to *Figure 3—source data 1* and *Supplementary file 3* for all statistical results). We observed significant genome-wide enrichment of breakpoints with the investigated transcriptional features (*Figure 3E* and F, 29% overlap with Pol2 stalling within gene regions, binomial test P=4.088e-07; 9% genome-wide overlap with convT, P=5.16e-07). This enrichment of breakpoints to convT and Pol2 stalling regions was significant across a wide range of transcriptional activity (refer to *Supplementary file 3*). Co-occurrence of breakpoints within a 1-kb window was used to distinguish non-recurrent (one breakpoint) and recurrent (more than one breakpoint) events (*Figure 3—source data 1*). Breakpoint recurrence was found to increase the overlap with both Pol2 stalling (*Figure 3E*)and convT (*Figure 3F*). The mean overlap observed in 1000-fold random sampling (grey bars) confirmed the specificity of the overlap (note that Pol2 stalling is analyzed from intragenic regions only). The breakpoints in Pol2 stalling sites were concordant with analysis using Pol2 ChIP-seq (by 78%) and they co-localized with both Ser2 and Ser5 phosphorylated forms of Pol2 complex (*Figure 3—figure supplement 2*). A schematic model summarizing the possible underlying mechanisms based on these results is shown in *Figure 3G*. The distinct integrated genomic profiles are collectively depicted at the *TPI1* loci, representing an SV hotspot with the highest number of NR-breakp in ETV6-RUNX1 cases (*Figure 3H*, see also *Figure 3—figure supplement 3* and *Figure 2A*). At the breakpoint region, both RLFS and convT are visible and overlap the elevated DRIP-seq signal measured from ES cells.

## Access to RAG cleavage sites increases at Pol2 stalling regions

Next, we focused on deciphering whether the transcriptional features associate with RAG off-targeting. We hypothesized that locally depleted nucleosomes around the Pol2 complex (*Bevington and Boyes, 2013*) may enhance access to RSS/RSS-like sequences. To this end, we retrieved DNAse hypersensitivity data from ENCODE (*The ENCODE Project Consortium, 2012*; see Materials and methods). DNAse-seq signal peaks were significantly wider when overlapping with Pol2 stalling sites (*Figure 4A*). A 876 bp (95% CI, 855–896) increase was observed in B-lymphoblastoid cells and 412 bp (95% CI, 395–429) in ES cells (Wilcoxon rank sum test P<2.2e-16 in both cell types, see also *Figure 3—source data 2*). This was reproducibly observed using peaks located within gene TSS, body or end regions (*Figure 4A*). We selected TSS regions with RSS motifs for closer examination and found that Pol2 stalling sites at these TSS were significantly wider than at other TSS (*Figure 4B*), with a difference of 259 bp (95% CI, 79–475 bp, Wilcoxon rank sum test P=0.0024). Thus, wide Pol2 stalling increases the likelihood of RSS motif occurrence in accessible chromatin. The width of stalling did not correlate positively (Pearson's correlation −0.11; 95% CI, −0.09 to −0.13) with the transcription level of the corresponding gene, indicating that stalling events, and not just active transcription, are important. We further analyzed the top 5% of widest Pol2 stalling regions by comparing them to widest peaks from DNAse hypersensitivity and ChIP for histone marks (see Materials and methods). The odds ratios for the overlap are visualized as a heatmap (see *Figure 4C*, OR>10 is shown in darkest color tone, refer to *Figure 4—source data 1* for more statistics). In addition to DNAse-seq and Pol2 ChIP peaks, the H3K4me3 was found among the top category, confirmed also by ChIP-seq data acquired from REH and Nalm6 cells (*Figure 4—source data 1*).

Next, the ETV6-RUNX1 R-breakp (335; 156 intragenic) were analysed for the genome-wide overlap with the transcriptional features. A 66% overlap was found with Pol2 stalling at intragenic regions (binomial test P<2.2e-16) and a 44% genome-wide overlap with convT (binomial test P<2.2e-16, see also *Figure 3—source data 1* for joint analysis across pre-B-ALL subtypes). The overlap with Pol2 stalling had high agreement between GRO-seq and ChIP-seq (*Figure 3—figure supplement 2*) and it increased at recurrent R-breakp (*Figure 4D*). In addition, overlap with convT (*Figure 4E*) was considerable (91%) at regions with 4 or more breakpoints. In comparison, regions with RLFS motifs or annotated TSSs showed less marked enrichment (up to 36%) (*Figure 4E*). Similar, as for NR-breakp, the significant overlap with transcriptional features was preserved at a wide range of expression levels (*Supplementary file 3*). A schematic model that links the obtained results with vulnerability to RAG cleavage is shown in *Figure 4F*. As in *Figure 3I*, the different profiles are depicted at the SV

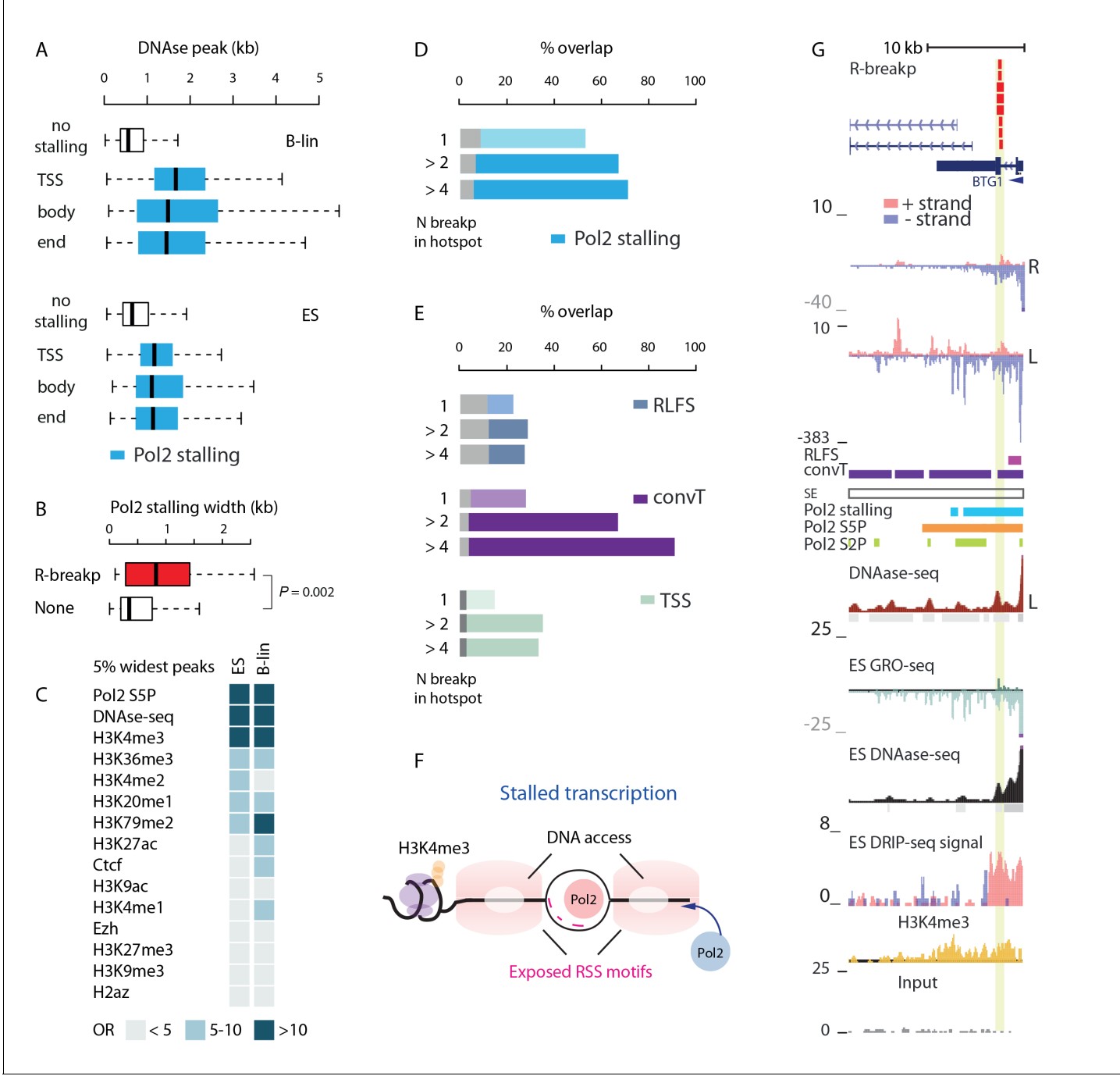

**Figure 4.** SV with RSS motifs localize to Pol2 stalling regions with broad open chromatin regions. (**A**) DNA access based on DNAse-seq peak width (GM12878 or H1 ES from ENCODE) is compared between regions with no Pol2 stalling (no color) and overlapping Pol2 stalling (light blue, cell-specific Pol2 stalling coordinates are listed in *Figure 2—source data 1*) at TSS, body and end region of transcripts (refer to *Figure 3—source data 2* for statistical analysis based on separate DNAse-seq replicates). (**B**) The TSS stalling width is compared between TSS harboring R-breakp and TSS with no breakpoints (Wilcoxon rank sum test P is indicated, TSS with R-breakp N = 38, TSS without breakpoints N = 11957, 95% CI for size difference 67–491 bp). (**C**) The 5% widest Pol2 stalling regions were overlapped with similarly defined widest peaks in different ChIP- and DNAse-seq data (refer to *Figure 4—source data 1* for details and all statistics). The odds-ratio (OR) for the overlap is visualized in color from discrete categories (<5; 5–10; >10, with darker color tones indicating higher OR). Pol2 S5P, DNAse-seq and H3K4me3 peaks had highest OR based on both B-lineage and ES cell data. **D** and **E**: The percentages of R-breakp overlapping Pol2 stalling (as in *Figure 3E*) or RLFS, convT and annotated TSS (as in *Figure 3F*) are shown as barplots, respectively. Overall, the recurrence was higher compared to NR-breakp and therefore two categories for recurrent R-breakp are shown (>2; >4). The overlap with convT reaches 91% at highly recurrent R-breakp hotspots (source data can be found in *Figure 2—source data 1*, S6 and statistics

*Figure 4 continued*

for genes binned by their transcription level in **Supplementary file 3**). (F) A schematic model illustrating how the transcriptional features may lead to the recruitment of RAG1 and RAG2 based on RSS-motif recognition and chromatin. Pol2 stalling associated with DNA accessibility and wide deposition of the H3K4me3 mark. (G) R-breakp hotspot with the highest recurrence (*BTG1* locus) is shown. B-lymphoblastoid and ES cell tracks from DNAse-seq and H3K4me3 from pre-B-ALL cells (Nalm6) represent signals with highest overlap to wide Pol2 stalling (other tracks as in **Figure 3H**, see also **Figure 4—figure supplement 1**).

The following source data and figure supplements are available for figure 4:

**Source data 1.** Overlap of wide Pol2 stalling regions with unusually wide peaks representing other chromatin features.
**Figure supplement 1.** Data from all signal tracks for regions displayed in **Figure 4**.
**Figure supplement 2.** GRO-seq signal profile at multiple clustered deletion regions.

hotspot with the highest number of R-breakp (**Figure 4G** *BTG1* locus, see also **Figure 4—figure supplement 1**). Further examples in **Figure 4—figure supplement 2** show RSS-dependent clustered deletions as defined in (**Papaemmanuil et al., 2014**). Overall, the presence of both convT and Pol2 stalling best characterized the recurrent ETV6-RUNX1 breakpoints with RSS motifs (101/148; compared to 20/70 without motif), with 90% (43/48, empirical P=0.002) co-occurrence at intragenic sites (see also **Supplementary file 4**).

## AID expression marks pre-B-ALL lacking common cytogenetic changes

To elucidate the potential for RAG and AID mediated genetic instability in leukemia blasts, we compared the expression of the genes *RAG1*, *RAG2* and *AICDA* across a transcriptome data set with 1382 pre-B-ALL patients (**Figure 5—source data 1**, **Figure 5**). Among samples with annotation of cytogenetic subtype (N = 1008), the ETV6-RUNX1 cases (N = 153) exhibited 10.8-fold higher median level of *RAG1* expression relative to other cases with annotated cytogenetic type (Wilcoxon rank sum test P<2.2e-16, 95% CI 8.6–13.6-fold, **Figure 5A**) and also high *RAG2* expression (**Figure 5B**). Moreover, *AICDA* expression was also detected in a specific subset of patients. It was highest in the 'other' group (N = 267) that does not carry recurrent fusion genes or karyotypic changes (**Figure 5C**, no statistical evaluation was performed as majority of signal values were below detection level of 4.2 in log2 scale). As comparison, we carried out unsupervised analysis of sample similarities based on the global gene expression profiles. To visualize these molecular subtypes in two dimensions, we utilized the t-Distributed Stochastic Neighbor Embedding (t-SNE) method (**van der Maaten and Hinton, 2008**) (see Materials and methods, refer to **Figure 5—source data 1** for coordinates). The t-SNE map places highly similar samples in close proximity. The discrete expression states (high; low; not detected) of *RAG1, RAG2* and *AICDA* were evident in distinct groups (**Figure 5D–F**, respectively, the annotated ALL subtypes are colored in **Figure 5G**). Upon further examination, high levels of *AICDA* expression were particularly prevalent in sample clusters that corresponded to high risk cases from two independent ALL datasets (hypergeometric test P=7.19e-47, **Figure 5H**, see **Supplementary file 5** for patient characteristics). The highest level of *AICDA* expression was presented by a relapsed ALL case, and the *RAG1* and *RAG2* expression levels were 3.09- and 1.93-fold increased at relapse, respectively. Based on the integrated patient profiles, the expression of AID and RAG is distinct in leukemia subtypes and clinical prognosis groups.

## Discussion

Next generation sequencing technologies have enabled the elucidation of mechanisms regulating transcription and the analysis of genetic alterations across different cancer genomes. Precursor leukemias are unique in that they often harbor SV and have relatively few mutations (**Roberts and Mullighan, 2015**). Recently, a functional role of transcription in genomic instability has begun to emerge (**Hatchi et al., 2015**; **Sollier et al., 2014**). The maturing lymphoid cells are vulnerable to off-target effects downstream of RAG and AID activity that is required for immune gene rearrangement (**Meng et al., 2014**; **Qian et al., 2014**, **Papaemmanuil et al., 2014**, **Swaminathan et al., 2015**). The

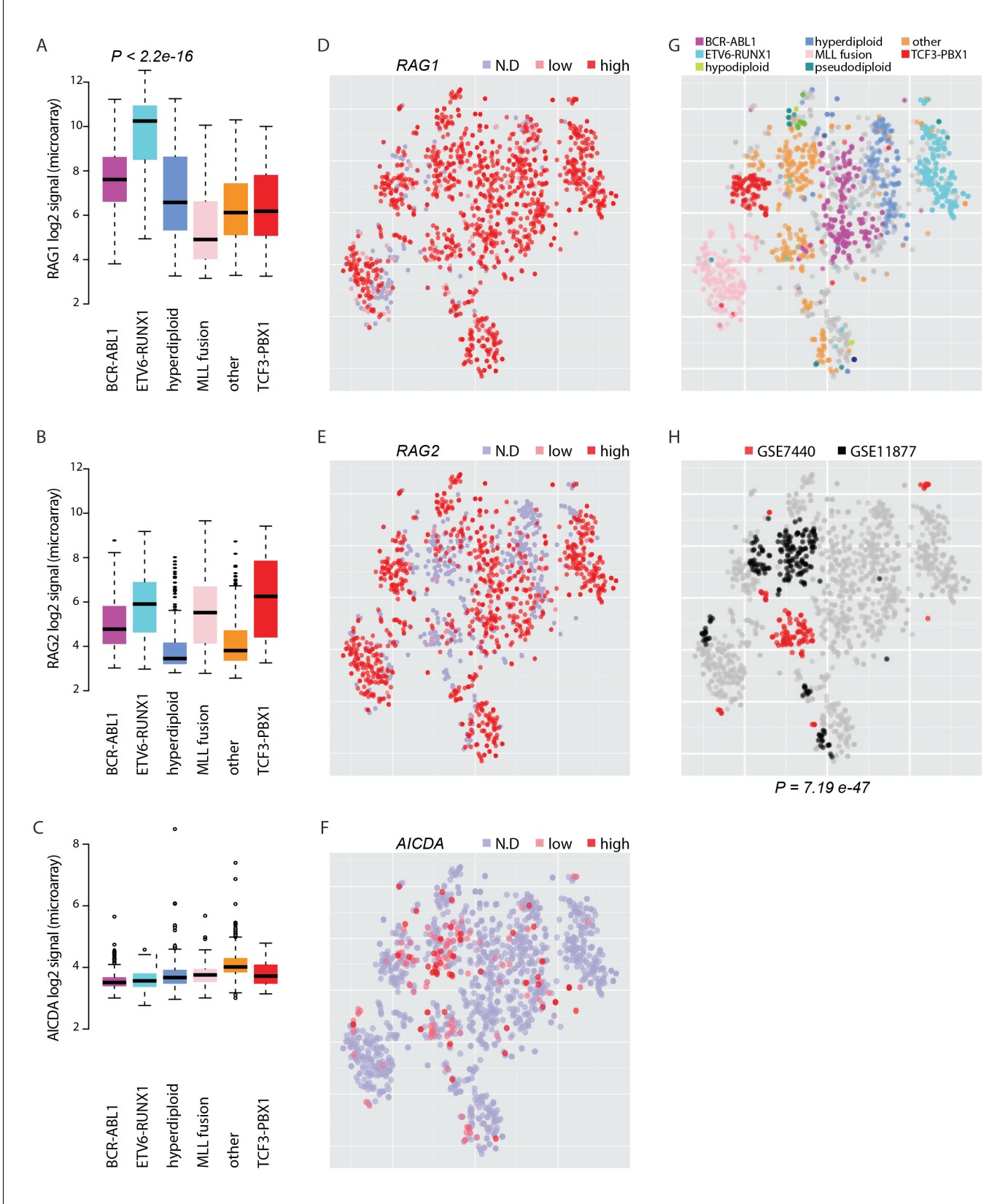

**Figure 5.** Expression of AID and RAG across molecular subtypes of leukemia. The log2 expression signal is summarized as boxplots for (**A**) *RAG1* (**B**) *RAG2* and (**C**) *AICDA* across the pre-B-ALL subtypes (*N* = 153 BCR-ABL1, *N* = 153 ETV6-RUNX1, *N* = 151 hyperdiploid, *N* = 198 MLL rearrangement, *Figure 5 continued on next page*

*Figure 5 continued*

N = 267 other, N = 82 TCF3-PBX1). Wilcoxon rank sum test p-value is indicated for differential *RAG1* expression in the ETV6-RUNX1 subtype (N = 153, patients with cytogenetic subtype information N = 1008) (in **A**). (**D–F**) Alternative representation of discrete expression states for *RAG1*, *RAG2*, and *AICDA*, respectively (red: high, pink: low, grey: not detected). The data points shown as a t-SNE map correspond to the full set of pre-B-ALL patient samples (N = 1382) (see also **Figure 5—source data 1**). Their relative positions are defined by the transcriptome similarity. The sample groups can be compared to annotated cytogenetic types, as colored on the same map in (**G, H**). The location of high-risk samples (N=295) from two independent studies is indicated in color on the same map (COG studies GSE7740 in red and GSE11877 in black, see also **Supplementary file 5**). Hypergeometric test p-value is indicated for enrichment of detected *AICDA* expression in the high risk studies (N = 112, refer to **Supplementary file 5** for population statistics).

The following source data is available for figure 5:

**Source data 1.** pre-B-ALL transcriptome samples.

present study represents a systematic investigation of SVs detected in acute pre-B-cell leukemia using WGS in the context of global transcriptional activity in leukemic cells. We identified specific transcriptional features, namely convergence of transcription and Pol2 stalling, as key factors underlying secondary genetic lesions frequently seen in precursor B leukemias.

Pol2 stalling and convT strongly associate with recurrent breakpoint sites across the genome and at gene loci implicated in leukemia such as *CDKN2A* and *PAX5* (**Sulong et al., 2009**). While protein-coding secondary hits required in disease progression have been recognized for some time, our integrative analysis identified several putative long non-coding RNAs and eRNAs, which merit further investigation. Earlier work has linked eRNAs generated from intragenic superenhancers with AID-mediated instability in lymphomas (**Meng et al., 2014**), proposing that convT leads to arrested transcription, in agreement with experimental evidence from yeast cells (**Prescott and Proudfoot, 2002**). Similarly, it has been shown that Pol2 stalling and R-loops expose ssDNA for AID targeting (**Huang et al., 2007**, **Pavri et al., 2010**, **Alt et al., 2013**). We show for the first time that leukemia breakpoints similarly display significant enrichment to enhancers overlapping convT. We further demonstrate a link between convT and elevated R-loop levels and Pol2 stalling on a genome-wide level, with evidence from normal and leukemic human cells. These mechanisms of transcription-coupled genetic instability, earlier implicated in lymphomas (**Pavri et al., 2010**; **Meng et al., 2014**; **Pefanis et al., 2014**) and breast cancer (**Hatchi et al., 2015**), therefore have relevance in multiple different cancer types.

Breakpoints carrying RSS-like recognition motifs for RAG1 showed high overlap with the vulnerable regions as defined by convT and Pol2 stalling. Therefore, we propose that also RAG1 access to its target sites is related to the fidelity of elongation. Previous studies investigating motif recognition and genome-wide binding profiles of RAGs have shed light on the mechanisms how this complex is recruited to DNA (**Bevington and Boyes, 2013**, **Teng et al., 2015**); however these studies have been carried out using normal cells or mouse models that limit their integration with patient WGS data. The chromatin mark H3K4me3 typically found at active promoters serves as a docking site for RAG2 (**Matthews et al., 2007**, **Teng et al., 2015**). RAG-mediated cleavage further requires recognition of RSS motifs by RAG1 (**Schatz and Swanson, 2011**). Our results revealed that TSS that carry breakpoints with an RSS motif differ from unaffected TSS by the presence of unusually wide Pol2 stalling. We show that Pol2 stalling sites, in general, have increased DNA accessibility. Further, the top 5% of widest stalling regions are characterized by unusually broad DNAse hypersensitive regions and H3K4me3 signal. Unique regulation of Pol2 pausing and elongation has been recognized to be related to broad H3K4me3 domains across a wide variety of cell types (**Benayoun et al., 2014**, **Scheidegger and Nechaev, 2016**). Together, these properties of Pol2 stalling sites may favor both the recognition and cleavage by the RAG complex.

In this study, we developed a genome-wide approach to capture Pol2 stalling events across gene bodies using change points analysis. This extends previous approaches to detect promoter-proximal pausing events (reviewed in **Adelman and Lis, 2012**) to analysis of slowing down of Pol2 within the full transcribed region. The feasibility of our approach was confirmed by high overlap of detected regions with RLFS rich regions that represent known structural obstacles to the progression of transcription (**Skourti-Stathaki et al., 2014a**; **Skourti-Stathaki et al., 2014b**). Furthermore, analysis of Pol2 stalling from Pol2 ChIP-seq profiles acquired in pre-B-ALL cells had high agreement with the

GRO-seq profiles. The slowing of Pol2 upon transition from initiation to elongation, measured by the Pol2 Ser5 phosphorylation, occurs at AID hypermutation sites within the IgH-V region (*Wang et al., 2014a*). We show that this type of Pol2 stalling had high overlap with leukemia breakpoints.

While RAG has a well-established role in pre-B cells, expression of AID represents a recently discovered threat for lymphoid precursor genome integrity. (*Swaminathan et al., 2015*) showed that infection-triggered attenuation of IL-7 receptor signaling led to strong AID expression, thus exposing pre-leukemic cells to additional off-targeting events. Moreover, a negative effect on patient survival and increased relapse frequency were observed in high *AICDA* expressing leukemia patients (*Swaminathan et al., 2015*). We found that high expression of *RAG1/2* or *AICDA* is markedly distinct between different subtypes of pre-B-ALL at the leukemia state. Prevalent *AICDA* expression was a distinguishing feature of high risk pre-B-ALL cases, in line with the previous data (*Swaminathan et al., 2015*). Furthermore, the molecular profiles of patients belonging to the cytogenetic subtype designated as 'other', had high similarity, placing them in close proximity on the t-SNE map. This genetically heterogeneous category of rare cytogenetic types had a distinct elevation in *AICDA* expression. Further investigation of the WGS profiles focusing on this patient category may shed light on whether *AICDA* expression could serve as a putative underlying factor that may spur the diversity of DNA lesions in these patients. Similarly, the over ten-fold higher *RAG1* expression could also be relevant for the prevalent development of leukemia carrying the ETV6-RUNX1 initiating fusion. The *RAG* locus is under complex regulation of local chromatin looping by SATB1 (*Hao et al., 2015*) that controls silencing and activating regulatory elements and was shown to directly control the elevated *RAG1* expression in mice. The enhancer activity in patient blast cells, as captured here in the nascent transcriptomes, will help understanding the regulation of such key loci in detail.

As more data on SV becomes available across cancers, further efforts should be made to elucidate the contribution of different complexes in transcription-coupled genomic instability and to develop strategies for dampening their levels and activity. Translation of these measures into clinical practice could impact treatment efficacy by decreasing clonal heterogeneity and relapse risk.

## Materials and methods

### GRO-seq samples

Primary bone marrow or blood samples from pediatric precursor B-ALL patients that represented different cytogenetic subtypes were used for GRO-seq assay (refer to *Supplementary file 1* for cytogenetic and blast count data). The study was approved by the Regional Ethics Committee in Pirkanmaa, Tampere, Finland (#R13109). The study was conducted according to the guidelines of the Declaration of Helsinki, and a written informed consent was received by the patient and/or guardians. In addition, three ALL cell lines (REH, Kopn-8 and Nalm-6) representing different genetic subtypes (ETV6-RUNX1 fusion, MLL rearrangement and 'other') were included to complement the dataset. REH (ACC-20), Nalm6 (ACC-128), and Kopn8 (ACC-552) cell lines were obtained from the Leibniz-Institut DSMZ-Deutsche Sammlung von Mikroorganismen und Zellkulturen GmbH (Germany). Mycoplasma status was defined negative by PCR (PCR Mycoplasma Test Kit I/C, PromoCell GmbH, Germany) for all cells. The cell lines were authenticated by PCR of known fusion genes: ETV6-RUNX1 in REH, MLL-MLLT1 in Kopn8, and the lack of recurrent fusions in Nalm6. In addition, the generated genome-wide results can be used in verification of cell line specific markers. We reasoned that a collection of samples that represent both primary blasts and cell lines of different cytogenetics types (and genetic complexity) would be ideal to capture the patterns of transcriptional activity in the lymphoid lineage and leukemic cells. Furthermore, 4–8 replicates were collected from a subset of samples to ensure reproducibility of the results (*Figure 2—figure supplement 2*). In cell culture studies, same cell lines with similar conditions are defined as biological replicates, as nuclei were extracted from temporally independent experiments. For nuclei extractions in co-culture experiments, same cell lines with different culture conditions but with same time points were processed simultaneously. For example, total of eight extractions from REH cells were performed ('Sample name' column), in six slightly different culture conditions ('Cell culture type' and 'Time point (h)' columns), and with two replicate samples collected for two conditions in independent experiments ('Biological replicate' column). There were no technical replicates in the sense that multiple nuclei extractions would have

been made from the same biological replicate. Patient samples (N = 7) collected represent different cytogenetic subtypes and were used as additional confirmation at individual gene loci: for most sub-types N = 1, except hyperdiploid N = 2; ETV6-RUNX1 is represented also by the REH cell line; and replicate samples were generated for Patient 1 that correspond to cultured and freshly isolated cells. (Re-analyzed GRO-seq data: lymphoblastoid data is from three donors; ESC data is from two independent experiments where several technical replicates were pooled). For cell culture conditions and further details, see *Supplementary file 1*. The nuclei isolation was performed as previously described (*Kaikkonen et al., 2013*), yielding ~1–5×10$^6$ nuclei per condition. The REH, Nalm6, lymphoblastoid and ES cell samples that represent very deeply sequenced data, were used in the genome-wide analysis (*Supplementary file 1*, GEO accession numbers for deposited pre-B-ALL data: GSE67519 and GSE67540).

## GRO-seq assay

Cells were suspended in 10 ml of swelling buffer (10 mM Tris-HCl, 2 mM MgCl2, 3 mM CaCl2 and 2 U/ml SUPERase Inhibitor [Thermofisher, Carlsbad, CA, USA] RNAse inhibitor) and let swell for 5 min. The cells were pelleted for 10 min at 400 × g and resuspended in 500 µl of swelling buffer supplemented with 10% glycerol. Subsequently, 500 µl of swelling buffer supplemented with 10% glycerol and 1% Igepal was added drop by drop to the cells while being vortexed gently. Nuclei were washed twice with 10 ml of swelling buffer supplemented with 0.5% Igepal and 10% glycerol, and once with 1 ml of freezing buffer containing 50 mM Tris-HCl pH 8.3, 40% glycerol, 5 mM MgCl2 and 0.1 mM EDTA. Nuclei were counted and centrifuged at 900 × g for 6 min and suspended to a concentration of 5 million nuclei per 100 µl of freezing buffer, snap-frozen and stored -80°C until run-on reactions. The nuclear run-on reaction buffer (NRO-RB; 496 mM KCl, 16.5 mM Tris-HCl, 8.25 mM MgCl2 and 1.65% Sarkosyl (Sigma-Aldrich, Steinheim, Germany) was pre-heated to 30°C. Then each ml of the NRO-RB was supplemented with 1.5 mM DTT, 750 µM ATP, 750 µM GTP, 4.5 µM CTP, 750 µM Br-UTP (Santa Cruz Biotechnology, Inc., Dallas, Texas, USA) and 33 µl of SUPERase Inhibitor (Thermofisher, Carlsbad, CA, USA). 50 µl of the supplemented NRO-RB was added to 100 µl of nuclei samples, thoroughly mixed and incubated for 5 min at 30°C. GRO-Seq libraries were subsequently prepared as previously described (*Kaikkonen et al., 2013*). Briefly, the run-on products were treated with DNAse I according to the manufacturer's instructions (TURBO DNA-free Kit, Thermofisher, Carlsbad, CA, USA), base hydrolysed (RNA fragmentation reagent, Thermofisher, Carlsbad, CA, USA), end-repaired and then immuno-purified using Br-UTP beads (Santa Cruz Biotechnology, Inc., Dallas, Texas, USA). Subsequently, a poly-A tailing reaction (PolyA polymerase, New England Biolabs, Ipswich, MA, USA) was performed according to manufacturer's instructions, followed by circularization and re-linearization. The cDNA templates were PCR amplified (Illumina barcoding) for 11–14 cycles and size selected to 180–300 bp length. The ready libraries were quantified (Qubit dsDNA HS Assay Kit on a Qubit fluorometer, Thermofisher, Carlsbad, CA, USA) and pooled for 50 bp single-end sequencing with Illumina Hi-Seq2000 (GeneCore, EMBL Heidelberg, Germany). GRO-Seq reads were trimmed using the HOMER v4.3 (http://homer.salk.edu/homer) software (homerTools trim) to remove A-stretches originating from the library preparation. From the resulting sequences, those shorter than 25 bp were discarded.

## ChIP-seq assay

ChIP-seq was performed using antibodies against the Ser2 and Ser5 phosphorylated forms of Pol2 and against the histone mark H3K4me3 in REH (N = 1) and Nalm6 cells (N = 2). Ser5 phosphorylation is present before the Pol2 is released to active elongation and it diminishes within the gene body and is greatly reduced downstream of the poly(A) site, where Ser2 phosphorylation is predominantly found (*Zhou et al., 2012*). For ChIP, 40 million cells were crosslinked with 1% formaldehyde for 10–15 mins. The reactions were quenched by adding glycine to a final concentration of 125 mM, and the cells were centrifuged and washed twice with ice-cold PBS. For ChIP 5 or 10 million cells were used (Pol2 or H3K4me3, respectively). Nuclei were extracted by washing cell pellet twice with 1 ml of MNase buffer (10 mM Tris ph 7.4, 10 mM NaCl, 5 mM MgCl2, 0.5% IGEPAL CA-630 [Sigma-Aldrich, Steinheim, Germany], 1x protease inhibitor cocktail [PIC, Roche, Basel, Switzerland], 1 mM PMSF [Thermofisher, Carlsbad, CA, USA]). Nuclei were spun down (1500 × g, +4°C, 5 min) and suspended into 90 µl of MNase buffer supplemented with 5 mM CaCl2 and 0.1% Triton-X. Different

amounts of MNase (0.5–20 U; #88216, Thermofisher, Carlsbad, CA, USA) was added to the nuclei in 10 μl volume and incubated at 37°C for 10 mins. To stop the reaction, 100 μl of 2x Lysis buffer was added to the reaction (1% SDS, 40 mM EDTA, 100 mM Tris-HCl pH 8.1) and samples were sonicated using Bioruptor (Diagenode) for 5 cycles (30 s - 30 s) to break the nuclei. The lysate was cleared by centrifugation and supernatant was diluted with RIPA buffer (for Pol2 antibodies, 1X PBS, 1% NP-40, 0.5% Sodium deoxycholate, 0.1% SDS, PIC) or dilution buffer (for H3K4me3; 20 mM Trix-HCl pH 7.4, 100 mM NaCl, 2 mM EDTA, 0.5% TritonX, PIC). The diluted lysate was pre-cleared by rotating for 2 h at 4°C with 60 μl 80% CL-4B sepharose slurry (GE Healthcare, UK). Before use, sepharose was washed twice with TE buffer, blocked for 1 hr min at room temperature with 0.5% BSA and 20 μg/ml glycogen in 1 ml TE buffer, washed twice with TE and brought up to the original volume with TE. The beads were discarded, and 1% of the supernatant were kept as ChIP input. The protein of interest was immunoprecipitated by rotating the supernatant with 3–5 μg antibody overnight at 4°C. Antibodies against Ser2P (cat# ab5095, RRID:AB_304749) and Ser5P (cat# ab5131, RRID: AB_449369) were purchased from Abcam (Cambridge, MA, USA). The Ab was captured using 25 μl blocked Protein G Sepharose 4 Fast Flow (GE Healthcare, UK) and rotating the sample for 2 hr at 4°C. Sepharose was blocked as CL-4B above, except that it was rotated overnight at 4°C. The beads were pelleted (1 min, 1000×g, 4°C) and the supernatant discarded. The beads used to bind Ser2P/ 5P Ab were washed five times with 5X LiCl IP wash buffer (100 mM Tris pH 7.5, 500 mM LiCl, 1% NP-40, 1% Sodium deoxycholate) and twice with TE in 0.45 μm filter cartridges (Ultrafree MC, Millipore, Bedford, MA, USA). The beads used to pull down H3K4me3 Ab were washed three times with wash buffer I (20 mM Tris/HCl pH 7.4, 150 mM NaCl, 0.1% SDS, 1% Triton X-100, 2 mM EDTA), twice with buffer II (20 mM Tris/HCl pH 7.4, 500 mM NaCl, 1% Triton X-100, 2 mM EDTA) and buffer III (10 mM Tris/HCl pH 7.4, 250 mM LiCl, 1% IGEPAL CA-630, 1% Na-deoxycholate, 1 mM EDTA), once with TE + 0.2% TritonX and twice with TE. Immunoprecipitated chromatin was eluted twice with 100 μl elution buffer (TE, 1% SDS). The NaCl concentration was adjusted to 300 mM with 5 M NaCl and crosslinks were reversed overnight at 65°C. The samples were sequentially incubated at 37°C for 2 h each with 0.33 mg/ml RNase A and 0.5 mg/ml proteinase K (both from Thermofisher, Carlsbad, CA, USA). The DNA was isolated using the ChIP DNA Clean & Concentrator (Zymo Research, Irvine, CA, USA) according to the manufacturer's instructions. Sequencing libraries were prepared from collected DNA by blunting, A-tailing, adaptor ligation as previously described (*Heinz et al., 2010*) using barcoded adapters (NextFlex, Bioo Scientific, Austin, TX, USA). Between the reactions, the DNA was purified using Sera-Mag SpeedBeads (Thermofisher, Carlsbad, CA, USA). Libraries were PCR-amplified for 15–16 cycles, size selected for 230–350bp fragments by gel extraction and single-end sequenced on a Hi-Seq 2000 (Illumina) for 50 cycles.

## Processing of GRO-seq, ChIP-seq, DRIP-seq and HiC sequencing reads

The GRO-seq data from lymphoblastoid cells (GSE39878, *Wang et al., 2014b*; GSE60456, *Core et al., 2014*), ES cells (GSE41009, *Sigova et al., 2013*), DRIP-seq data from ES cells (GSE45530, *Ginno et al., 2013*) and HiC data from human lymphoblastoid GM12878 cells (GSM1551571, GSM1551572, GSM1551574, GSM1551575; *Rao et al., 2014*) were downloaded from SRA (raw reads) and processed similarly as the new samples: reads were quality controlled and subsequently aligned to the human hg19 reference genome version. Specifically, the quality of raw sequencing reads was confirmed using the FastQC tool (http://www.bioinformatics.babraham.ac.uk/ projects/fastqc/) and subsequently bases with poor quality scores were trimmed (requiring a minimum 97% of all bases in one read to have a min phred quality score of 10) using the FastX toolkit (http://hannonlab.cshl.edu/fastx_toolkit/). Samples sequenced on multiple lanes were pooled after quality control. Read stacks were collapsed from ChIP-seq files using fastx (collapse). The Bowtie software (bowtie-0.12.9v0.1.x) (*Langmead et al., 2009*) was used for aligning the GRO-seq, ChIP-seq and DRIP-seq reads to the human genome (version hg19). Up to two mismatches and up to three locations were accepted and the best alignment was reported for each read. For the GRO-seq reads this step was preceded by removing reads mapping to rRNA regions (AbundantSequences as annotated by iGenomes) and discarding reads overlapping with so-called blacklisted regions that represent unusual low or high mappability as defined by ENCODE, ribosomal and small nucleolar RNA (snoRNA) loci from ENCODE and further manually curated for the human genome (bed file with sequences is provided in *Supplementary file 6*).

## HiC

Reads from paired-end sequencing were separately filtered and aligned to the genome using bowtie. The reads were checked for MboI restriction sites before doing the alignments and the sequences after GATC sites were trimmed out to improve mappability. The HOMER v4.3 (http://homer.salk.edu/homer) software was used in further processing of HiC-data. Paired-end reads were connected and read pairs with exact same ends were only considered once and read pairs were removed if they were separated by less than 1.5× the estimated sequencing insert length to remove likely continuous genomic fragments or re-ligation events. Paired-end reads originating from regions of unusually high tag density were left out by removing reads from 10 kb regions that contain more than five times the average number of reads. Background model for normalization of HiC-data was generated with 50 kb resolution. The topological domains were identified using the HOMER command' findHiCDomains.pl' using a resolution of 50 kb. This analysis is based on a statistic referred to as the 'directionality index', which describes the tendency of a given position to interact with either the chromatin upstream or downstream from its current position.

## GRO-seq

Combined tagDirectories from GRO-seq samples were made by pooling the sequencing data for each cell and sample type with fragment length set to 75. The findPeaks.pl program in the The HOMER v4.3 software (http://homer.salk.edu/homer) was used to identify *de novo* transcripts from GRO-seq data using pooled sequencing reads per sample type. Deeply sequenced REH, Nalm6 and lymphoblastoid cells were used to define signal features in B-cell lineage and separate analysis was carried out for ES cells (see *Supplementary file 1*). Gaps were allowed at non-mappable regions (-style groseq -uniqmap).

## ChIP-seq

Peaks were identified using findPeaks (-style histone –size 1000).

## Signal tracks

BedGraph and bigWig files were generated with reads in each sequencing experiment normalized to a total of $10^7$ mapped reads. The bigWig files were further converted to track hubs and visualized as strand-specific, overlaid MultiTracks as a custom Track Hub in the UCSC Genome browser.

## Genomic regions used in analyses

The hg19 genome version from UCSC (available from iGenomes) was used to specify chromosome lengths in the analysis. The gene annotations from Refseq and UCSC known gene tables were retrieved using the UCSC Table Browser (hg19, GRCh37 Genome Reference Consortium Human Reference 37 (GCA_000001405.1)). Unique transcript coordinates were used in analysis, that is, any transcripts sharing the same start and end coordinate were considered together. The TSS regions were defined as +/- 1 kb regions around the annotated start coordinate. Only transcripts mapping to canonical chromosomes were kept, also those on chrM were removed.

## Enhancers

Super-enhancer coordinates from CD19+, CD20+ and HSC cells were obtained (*Hnisz et al., 2013*) and merged for visualization of tracks. De novo enhancer detection was performed from the deeply sequenced REH, Nalm6 and lymphoblastoid cells based on the transcript identification result. Transcripts with length <15 kb and the characteristic bidirectionality or co-localization with enhancer locations defined using DNAse and chromatin marker data were used to distinguish eRNAs (see *Figure 2—source data 1* for data).

## Analysis of SV in context of chromosome subregions

TADs reflect the three dimensional structure of chromatin, forming natural boundaries that divide the chromosomes into sub-regions. To identify TADs with highest frequency of breakpoints, HiC-data analysis was performed using HOMER 4.3. As our goal is to generate a natural division of the genome into sub-regions that are relevant in context of transcriptional regulation, this approach is superior to arbitrarily assigning sub-regions based on fixed windows. The pre-B-ALL breakpoints and

annotation data (*Andersson et al., 2015*; *Holmfeldt et al., 2013*; *Papaemmanuil et al., 2014*; *Paulsson et al., 2015*) were analyzed in context of TADs. Specifically, TADs were overlapped with subtype-specific breakpoints (*Figure 1—source data 1* presents TADs sorted based on the count of breakpoints). Subsequently, TADs with breakpoints were divided into quartiles based on breakpoint frequency per bp to analyze enrichment of feature overlap that exceeds the genomic background level. To obtain the total transcribed area width within each TAD, the TAD coordinates were overlapped with the detected GRO-seq transcripts (bedtools intersect -wao). The combined SV data represents in total 1680 breakpoints and is the most comprehensive collection of pre-B-ALL SV that we are aware of.

## Chromatin segmentation data

BroadChromHMM chromatin segmentations were obtained from GM12878 and H1 ES cells including the following segment types: 1_Active_Promoter, 2_Weak_Promoter, 3_Poised_Promoter, 4_Strong_Enhancer, 5_Strong_Enhancer, 6_Weak_Enhancer, 7_Weak_Enhancer, 8_Insulator, 9_Txn_Transition, 10_Txn_Elongation, 11_Weak_Txn, 12_Repressed", 13_Heterochrom/lo, 14_Repetitive/CNV, 15_Repetitive/CNV. The sizes of the segments of each type were used to calculate the total span from the genome. Each segment type was then overlapped with a combined bed file specifying convT and Pol2 stalling regions. Overlapping and non-overlapping pieces were returned and analyzed separately (bedtools intersect, followed by bedtools subtract with the overlapping pieces given as parameter b).

## Distinguishing breakpoints based on RSS-like motifs or recurrence

Two types of breakpoints were distinguished based on RSS motif annotation to result in the following region assignment: regions containing a consensus RSS/heptamer sequence motif were used to categorize co-localized breakpoints as putative RSS-dependent breakpoints (R-breakp: 447 in total, 335 in the ETV6-RUNX1 subtype), while regions devoid of recognition sequence were used to classify RSS-independent lesions (NR-breakp: 938 in total, 416 in the ETV6-RUNX1 subtype). Regions harboring unresolved breakpoints were left out from majority of analysis performed (285 regions harboring 295 breakpoints in the ETV6-RUNX1 subtype that were mainly isolated and non-recurrent). The RSS assignment for other breakpoints was obtained in the following way: the resolved breakpoints were extended to both sides by 10 bp, resulting in a 21 bp region. The MEME analysis in *Papaemmanuil et al. 2014* for 708 resolved breakpoints from ETV6-RUNX1 patients was replicated and comparable sequence logos to that reported previously were obtained and used to annotate RSS status. A p-value cut-off of 0.003 was chosen for the MEME motif scanning based on FIMO analysis of the ETV6-RUNX1 data.

To evaluate recurrence, the breakpoint ends at 1 kb distance were stitched together to form regions (each with at least one breakpoint, see *Figure 3—source data 1*), annotating the number of breakpoints inside (BEDTools mergeBed –d 1000 –n). Overlap of breakpoint regions with RLFS, TSS, convT and Pol2 stalling regions were obtained using BEDTools with 1 kb window. The overlap frequencies were compared to random sampling of similarly sized genomic regions. Further comparisons were performed separating recurrent (>1 breakpoint per stitched region) and non-recurrent regions, and with increasing the cut-off for the number of breakpoint events per stitched region. The same was repeated for breakpoints within genes binned into four categories based on their transcription level. The transcript regions were quantified using data from REH, Nalm6, and lymphoblastoid cells, and normalized by RPKM. The maximum expression value was to divide transcripts into quartiles based on the expression level.

## Signal feature analysis

The visual examination of SV sites served as the first step to define transcriptional features of potential relevance. This motivated the analysis of regions with overlapping transcription from both strands (convT) and local elevations in the signal (Pol2 stalling), with detailed definitions given below. For genome-wide analysis of signal feature overlap with SV, feature tracks from several samples were combined. This approach was deemed most appropriate to address the dynamic nature of transcriptional activity and to avoid missing regions that due to high recurrence of SV may be deleted in subset of leukemic cells studied.

## ConvT

ConvT regions were identified as transcripts that overlap on opposite strands by at least 100 bp (as in *Meng et al., 2014*). Subsequently, a combined bed track was created for the leukemic and lymphoblastoid samples using bedTools mergeBed command (-d 0). The data from ES cells (GSE41009) served as an independent control. The level of convT was quantified using the HOMER program analyzeRNA.pl from both strands separately and normalized by region size. The minimum value obtained per region (comparing + and - strands) was assigned as convT level.

## Pol2 stalling

Change-point detection is the mathematical problem of finding abrupt changes in a signal, typically applied in context of time series (*Killick et al., 2012*). Both approximate and exact methods exist for estimating the point at which the statistical properties of a sequence of observations change. The analysis of changepoints in the signal mean was carried out using functions implemented in the R package 'changepoint' (*Killick and Eckley, 2014*). An exact method with favorable computational cost was recently introduced in context of time-series data (*Killick et al., 2012*). This method called PELT was selected to detect the changepoints from scaled (zero mean, equal variance) signal profiles calculated at 50-bp resolution (generated bedGraph files are available under the GEO accession GSE67540). The analysis was performed separately for each gene, using Bayesian Information Criterion as a penalty term with the changepoint counted as a parameter (function cpt.mean with parameters penalty = 'BIC1', method ='PELT'). The input dataset representing primary transcription activity at gene loci was generated by overlapping the GRO-seq signal file strand-specifically with transcript coordinates from UCSC and Refseq (see genomic regions used). The analysis only considered regions with annotation match (in minimum 5% of identified transcript covered by annotation; a minimum of 50% overlap with the identified transcript; annotated and detected starts do not differ more than 10 kb). In order to define Pol2 stalling sites, the signal level between changepoints were compared to the median across the whole gene, and intervals above 90% quantile were reported as stalled. For ChIP-seq, this cut-off was relaxed to 80% due to higher background signal. Notice also that there is no strand information based on ChIP-seq. The analysis was carried out separately for each of the deeply sequenced (REH, Nalm6 and lymphoblastoid) GRO-seq datasets and ChIP-seq replicates and subsequently merged to one bed file specifying stalled region coordinates (bedTools merge –d 100). The ES GRO-seq dataset GSE41009 was processed similarly and used as an independent control. To study whether there was a relationship between the stalled region size overlapping TSS regions and R-breakp frequency, the following intersects were calculated using BEDTools (intersectBed -wa | uniq): (i) overlap of Pol2 stalling sites and TSS regions harboring R-breakp and (ii) overlap of Pol2 stalling sites and TSS regions not harboring R- or NR-breakp. Subsequently, the sizes of Pol2 stalling sites in each coordinate file were calculated and the Wilcoxon rank sum test used for evaluating statistical significance for the difference in Pol2 stalling width. Secondly, top 5% widest Pol2 stalling sites were identified and compared to top 5% widest peaks from ChIP-seq and DNAse-seq profiles (see below).

## Signal comparison at gene regions

The HOMER command annotatePeaks.pl was used to create a transcriptional profile of active genes (RPKM > 0.5) in ES, lymphoblastoid and REH cells by scaling the histogram to each region (i.e 0–100%) using a bin size of 100. RLFS motif density was calculated across genes with the BEDtools coverage tool. A density plot representing RLFS frequency across gene regions was then obtained as above.

## DRIP-seq and RLFS motif data for R-loop detection

Data from replicate DRIP-seq experiments with two different restriction enzyme digestions (GSE45530) were used in the analysis. Log2 signal levels were quantified using HOMER at TSS regions. Statistical significance was estimated separately for the two different restriction enzyme digestions. RLFS motif search was performed using the software QmRLFS-finder that predicts R-loop forming sequences based on structural models of known sequences (*Jenjaroenpun et al., 2015*). The fasta input file was generated by extracting DNA sequences based on the hg19 genome version.

## DNAse-seq data and additional ChIP-seq data to characterize wide Pol2 stalling events

The DNAse-seq peaks were first overlapped with the Pol2 stalling regions detected based on the GRO-seq signal. Only peaks with a score above 15 were considered. The width of the overlapping peaks was then compared to the width of peaks with no overlap using the Wilcoxon rank sum test. Next, top 5% widest peaks were obtained from the DNAse-seq and ChIP-seq data (refer to *Figure 4—source data 1*). The overlap with 5% widest Pol2 stalling regions was subsequently evaluated using the BEDTools fisher tool.

## Transcriptome data

Gene expression data from pre-B-ALL studies was combined from microarray datasets retrieved from the NCBI GEO database as part of a data collection representing both healthy and malignant samples hybridized to hgu133Plus2 genome-wide microarrays (in preparation for submission). In total, 1382 pre-B-ALL samples were included. Probe-level quality control was performed to exclude samples with very high difference in data location or distribution as measured by median and inter-quartile range of raw probe intensities. Samples that passed this filtering were processed using the RMA probe summarization algorithm with probe mapping to Entrez Gene IDs (from BrainArray version 18.0.0, ENTREZG), followed by bias correction using the R package 'bias'. The Barnes-Hut-SNE algorithm (computationally faster approximation of t-SNE) implementation from the R package 'Rtsne' (*Krijthe, 2015*) was used to discover near-optimal representation of sample distances in two dimensions (using parameter values perplexity 30 and theta 0.5) using 15% genes with highest variance. The t-SNE method belongs to dimensionality reduction methods that include also traditional methods such as Principal Component Analysis. The main objective of the method is to accurately place highly similar samples (here based on the high-dimensional gene expression profile) to close proximities in lower dimensions. The result can be visualized in two-dimensions as a scatter plot that allows observing sample groups based on the molecular profiles. According to our experience, this method provides better separation between sample groups compared to more traditional methods for large heterogeneous sample collections. To identify whether a given gene was expressed or unexpressed in a sample, a Gaussian finite mixture model (testing equal and variable variance models, best fit chosen by BIC) was fitted by expectation-maximization algorithm to the probe signals (R package 'mclust', version 4.3, *Fraley and Raftery, 2002*).

## Statistical tests

Statistical significance was estimated using several tests to ensure reliability, including tests that rely on assumptions about data distributions and empirical tests that rely on randomization of data points. The statistical tests used, exact values of N, definitions of center and dispersion and precision measures are indicated in Results, in the respective supplementary tables or figure legends.

### Binomial test

Test for independent random trials with binary (success/failure) outcome, with replacement. This test was used to assess the statistical significance of observing breakpoint events overlapping a transcriptional feature (Pol2 stalling or convT). Success in population was defined using 1 kb windows across the genome. The windows overlapping the studied feature was divided by the total number of 1 kb windows analyzed. E.g. in the Pol2 stalling analysis, the total number of windows overlapping Pol2 stalling regions divided by this number of 1 kb windows within gene coordinates (included to the input for the change point analysis), define probability of success.

### Hypergeometric test

Test for independent random trials with binary (success/failure) outcome, without replacement. This test was used to assess the statistical significance of observing greater than or equal overlap frequency between breakpoints and an annotated set of genomic regions. E.g. to test for enrichment of breakpoints inside convT-positive enhancers, convT-positive enhancers with breakpoints define sample success; all enhancers with breakpoints population success; and sample taken is all convT-positive enhancers (from the population of all enhancers). The related Fisher's test (implemented in BEDTools fisher) was used to obtain similar statistics with odds ratios.

## Wilcoxon rank sum test (Mann-Whitney test)

A nonparametric two-sided Wilcox test was performed to estimate whether two samples (continuous values, unknown distribution) come from the same population (R function wilcox.test). This test was applied to quantified signal levels compared between categories.

## Random sampling

This test can be used to obtain an empirical estimate of random overlap frequencies. The sampling was performed 1000-fold within the same genomic context as used in the analysis. To estimate the significance of overlap between stitched breakpoint regions with e.g. convT regions, the stitched regions were allocated random genomic coordinates, thus preserving the size distribution and breakpoint event frequencies within stitched regions. The observed random region overlap was used as the empirical p-value estimate. Further, the z-test was used to evaluate whether there was evidence to reject the null hypothesis that the observed feature overlap value would belong to the empirical distribution obtained.

# Acknowledgements

We would like to thank Ville Hautamäki for comments on signal analysis methods and the EMBL Gene Core sequencing team for the sequencing service provided. The work was supported by grants from the Emil Aaltonen Foundation, Jane and Aatos Erkko Foundation, Finnish Cancer Foundation, Academy of Finland, Sigrid Juselius Foundation,Finnish Cultural Foundation, Paulo Foundation, Foundation for Pediatric Research, the Competitive State Research Financing of the Expert Responsibility area of Tampere University Hospital, University of Tampere and University of Eastern Finland.

# Additional information

### Funding

| Funder | Grant reference number | Author |
|---|---|---|
| Suomen Kulttuurirahasto | 00150214 | Merja Heinäniemi<br>Olli Lohi |
| Itä-Suomen Yliopisto | | Merja Heinäniemi |
| The Finnish Cancer Foundation | | Merja Heinäniemi |
| Emil Aaltosen Säätiö | | Merja Heinäniemi |
| Suomen Akatemia | 276634 | Merja Heinäniemi |
| Tampereen Yliopisto | | Susanna Teppo<br>Saara Laukkanen<br>Thomas Liuksiala<br>Olli Lohi |
| Sigrid Juselius Foundation | | Minna U Kaikkonen |
| Suomen Akatemia | 277816 | Olli Lohi |
| Jane ja Aatos Erkon Säätiö | | Olli Lohi |
| Paulo Foundation | | Olli Lohi |
| Lastentautien Tutkimussäätiö | | Olli Lohi |
| Competitive State Research Financing of the Expert Responsibility area of Tampere University Hospital | | Olli Lohi |

The funders had no role in study design, data collection and interpretation, or the decision to submit the work for publication.

## Author contributions

MH, ST, MUK, Conception and design, Acquisition of data, Analysis and interpretation of data, Drafting or revising the article; TV, OL, Conception and design, Analysis and interpretation of data, Drafting or revising the article; MB-L, JM, HN, TL, Acquisition of data, Analysis and interpretation of data, Drafting or revising the article; VZ, Analysis and interpretation of data, Drafting or revising the article; SL, KT, Acquisition of data, Drafting or revising the article

## Author ORCIDs

Merja Heinäniemi, http://orcid.org/0000-0001-6190-3439
Susanna Teppo, http://orcid.org/0000-0003-2569-8030
Olli Lohi, http://orcid.org/0000-0001-9195-0797

## Ethics

Human subjects: The study was approved by the Regional Ethics Committee in Pirkanmaa, Tampere, Finland (#R13109). The study was conducted according to the guidelines of the Declaration of Helsinki, and a written informed consent was received by the patient and/or guardians.

# Additional files

## Supplementary files

• Supplementary file 1. GRO-seq sample summary. Description of the patient and cell line GRO-seq samples used in the analysis, including the cell culture conditions, replicate information and the total number of pooled sequencing reads obtained after quality filtering and alignment. A more detailed table for cultured samples with replicate information and accession codes is provided at the bottom. Sample accession codes for already published and re-analyzed GRO-seq data, and additional GRO-seq data displayed in *Figure 1—figure supplement 1* are listed in worksheet 2.

• Supplementary file 2. Genomic coordinates for regions displayed. The coordinates of example gene regions displayed in the main and supplementary figures are listed (hg19 human genome version).

• Supplementary file 3. Breakpoint hotspot analysis for genes binned by the transcription level. Hypergeometric test statistics for genes stratified by expression level. Breakpoint overlap with transcriptional features was tested within the binned intragenic regions. Data for ETV6-RUNX1 subtype and all pre-B-ALL subtypes are shown as separate worksheets. Related to *Figures 3* and *4*.

• Supplementary file 4. Intragenic recurrent SV in ETV6-RUNX1 patients with overlap to vulnerable regions. The patient and region identifiers for recurrent intragenic SV in ETV6-RUNX1 patients are listed, reporting separately those co-localized with Pol2 stalling or convT regions.

• Supplementary file 5. Clinical data for patients with high *AICDA* expression. Study description, sample identifier, cytogenetic group, age and dataset identifier are listed for the patients within high *AICDA* expression level. Statistical analysis testing enrichment of detected AICDA expression in high risk studies is summarized in worksheet 2.

• Supplementary file 6. Custom blacklisted genomic regions. Blacklisted regions discarded from the analysis that were deemed to represent low-mappability, rRNA and snoRNA loci based on GRO-seq signal. Coordinates refer to the hg19 human genome version.

## Major datasets

The following datasets were generated:

| Author(s) | Year | Dataset title | Dataset URL | Database, license, and accessibility information |
|---|---|---|---|---|
| Heinäniemi M, Teppo S, Kaikkonen MU, Bouvy-Liivrand M, Lohi O | 2015 | ALL cells | http://www.ncbi.nlm.nih.gov/geo/query/acc.cgi?acc=GSE67540 | Publicly available at NCBI Gene Expression Omnibus (accession no: GSE67540) |
| Heinäniemi M, Teppo S, Lohi O | 2015 | Genome-wide mapping of TEL-AML1 targets in acute leukemia | http://www.ncbi.nlm.nih.gov/geo/query/acc.cgi?acc=GSE67519 | Publicly available at NCBI Gene Expression Omnibus (accession no: GSE67519) |

The following previously published datasets were used:

| Author(s) | Year | Dataset title | Dataset URL | Database, license, and accessibility information |
|---|---|---|---|---|
| Wang IX, Core LJ, Kwak H, Brady L, Bruzel A, McDaniel L, Richards AL, Wu M, Grunseich C, Lis JT, Cheung VG | 2014 | RNA-DNA DIFFERENCES IN NASCENT RNA | http://www.ncbi.nlm.nih.gov/geo/query/acc.cgi?acc=GSE39878 | Publicly available at NCBI Gene Expression Omnibus (accession no: GSE39878) |
| Core LJ, Martins AL, Danko CG, Waters CT, Siepel A, Lis JT | 2014 | Analysis of transcription start sites from nascent RNA identifies a unified architecture of initiation at mammalian promoters and enhancers | http://www.ncbi.nlm.nih.gov/geo/query/acc.cgi?acc=GSE60456 | Publicly available at NCBI Gene Expression Omnibus (accession no: GSE60456) |
| Sigova AA, Mullen AC, Molinie B, Gupta S, Orlando DA, Guenther MG, Almada AE, Lin C, Sharp PA, Giallourakis CC, Young RA | 2013 | Divergent transcription of lncRNA/mRNA gene pairs in embryonic stem cells | http://www.ncbi.nlm.nih.gov/geo/query/acc.cgi?acc=GSE41009 | Publicly available at NCBI Gene Expression Omnibus (accession no: GSE41009) |
| Ginno PA, Lim YW, Lott PL, Korf I, Chédin F | 2013 | DNA-RNA Immunoprecipitation sequencing (DRIP-seq) of human NT2 cells | http://www.ncbi.nlm.nih.gov/geo/query/acc.cgi?acc=GSE45530 | Publicly available at NCBI Gene Expression Omnibus (accession no: GSE45530) |
| Sanborn AL, Rao SS, Huang SC, Durand NC, Huntley MH, Jewett AI, Bochkov ID, Chinnappan D, Cutkosky A, Li J, Geeting KP, Gnirke A, Melnikov A, McKenna D, Stamenova EK, Lander ES, Aiden EL | 2014 | A three-dimensional map of the human genome at kilobase resolution reveals prinicples of chromatin looping | http://www.ncbi.nlm.nih.gov/geo/query/acc.cgi?acc=GSE63525 | Publicly available at NCBI Gene Expression Omnibus (accession no: GSE63525) |
| Sandstrom R | 2011 | DNaseI Hypersensitivity by Digital DNaseI from ENCODE/University of Washington | http://www.ncbi.nlm.nih.gov/geo/query/acc.cgi?acc=GSE29692 | Publicly available at NCBI Gene Expression Omnibus (accession no: GSE29692) |
| Shoresh N | 2011 | Histone Modifications by ChIP-seq from ENCODE/Broad Institute | http://www.ncbi.nlm.nih.gov/geo/query/acc.cgi?acc=GSE29611 | Publicly available at NCBI Gene Expression Omnibus (accession no: GSE29611) |

| Sandstrom R | | 2011 | CTCF Binding Sites by ChIP-seq from ENCODE/University of Washington | http://www.ncbi.nlm.nih.gov/geo/query/acc.cgi?acc=GSE30263 | Publicly available at NCBI Gene Expression Omnibus (accession no: GSE30263) |
| Myers R, Pauli F | | 2011 | Transcription Factor Binding Sites by ChIP-seq from ENCODE/HAIB | http://www.ncbi.nlm.nih.gov/geo/query/acc.cgi?acc=GSE32465 | Publicly available at NCBI Gene Expression Omnibus (accession no: GSE32465) |

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
