## [Decision Letter]

Thank you for submitting your work entitled "Transcription-coupled genetic instability marks acute leukemia structural variation hotspots" for consideration by *eLife*. Your article has been reviewed by two peer reviewers, and the evaluation has been overseen by a Reviewing Editor and Jessica Tyler as the Senior Editor.

The reviewers have discussed the reviews with one another and the Reviewing Editor has drafted this decision to help you prepare a revised submission.

Summary:

A series of recent studies demonstrated that genetic diversification during adaptive immune responses of B cells comes with an increased risk of malignant transformation, i.e. when AID, RAG1 and RAG2 target non-immunoglobulin genes. Based on integrative analyses of transcriptional activity (GRO-seq) and genetic lesions (whole genome sequencing), Lohi and colleagues propose a novel scenario to explain how AID and RAG1/RAG2 can be aberrantly targeted to Non-Ig sites and thereby cause genetic lesions that drive malignant transformation.

Essential revisions:

1) The scenario in Figure 6 is potentially interesting but goes far beyond what is supported by the actual data presented in the study. The Discussion section dilutes the main findings by additional speculation. A more detailed discussion of their actual data (instead of what could be gleaned from hypothetical future experiments) would be helpful. There is a lot of speculation on the Results section and in particular around findings that build up to this mechanism and it is not clear how the data/analysis supports these observations.

2) The manuscript is of potential interest to a broader audience but mainly written for expert readers in computational biology. To make this work accessible to a wider group of scientists that are potentially interested in the topic, it would seem to be necessary to explain the rationale for the use of certain methods and techniques in some meaningful detail. For instance, the use of t-SNE blots (as for CyTOF) to correlate AID, RAG1 and RAG2 expression with cytogenetic subtypes is of interest, but not discussed what exactly can be seen in the diagrams and why this method was chosen.

3) Expression of AID and RAG1/RAG2 "markedly distinct between pre-B ALL subtypes": The measurements were performed in fully established ALL clones. The interpretation that different expression levels of AID and RAG1/RAG2 point to a different role of these enzymes in different cytogenetic subtypes is likely incorrect. As in B cell lymphoma, AID and RAG1/RAG2 act together in a multi-step process of clonal evolution *towards* full transformation. In the fully established leukemia, secondary genetic events and consequences of oncogenic signaling may alter expression levels and obscure the role of AID and RAG1/RAG2 in these leukemia subsets. In addition, the significance of AID and RAG1/RAG2 expression only measured at the mRNA level is unclear. Western blot analyses should be performed if authors feel strongly about documenting different expression levels.

4) Do the proposed transcriptional features of instability at SV-sites coincide with cryptic RSS or minimal RAG1/RAG2 substrates? This would add to mechanistic plausibility of their scenario.

5) For the TAD Analysis: Would TAD definition change if a lymphoid origin cell lines or tissue types were used? The authors divide TADS in quartiles on the basis of number of breakpoints within TADs and study how this correlates with% study of convT at these loci. The size range and density of the TADs can vary greatly (T2). The authors should ensure that TAD size does not confound this analysis.

6) Correlation of 'transcriptional features' such as prevalence of POL2 stalling, DNA sequences that are susceptible to R-LOOP formation and also convergent transcription, with regions that are frequently rearranged in ALL, and in regions with high prevalence of RSS.

7) Whilst there is a notable enrichment against other active promoters for example the 'width' of the 'stalling' region, although what this really represents, as defined from GRO-seq data, is the high density of active RNA polymerases, so is it possible that these represent regions of active transcription.

8) With previous reports showing enrichment of genomic rearrangements at active promoters and enhancers it is not clear whether the observations here are a consequence of transcriptional activity in these regions or an enriched mechanism underpinning the regions that are frequently targeted. The authors should present a global analysis – see statistical review section.

9) A major limitation of the analysis here is that there are not overlapping datasets from the same samples. As the authors have access to direct primary patient cells and cell lines representative of B-ALL their interpretations of the data would significantly benefit from performing RNAseq, POL2 chip, and potentially MNase-seq to support their observations (Pol2 stalling and convergent transcription).

10) Overall there is a clear narrative issue, the dataset is complex and the analysis is not clear nor comprehensive – making it rather hard for the reviewer and in time a reader to comprehensively evaluate the analytical approaches used, as well as, the evaluate the interpretation of the findings, which often render conclusions as facts by citing other papers rather than being supported by the data in itself. It would be very helpful if the authors included a supplemental figure containing a flow chart describing what datasets where put together and which subsets of data were used for which analysis.

11) Results should be interpreted on the basis of their analysis. Conclusions that account or include findings from the literature should be placed in the Discussion section.

12) It would be useful if the authors performed a global analysis- accounting for chromatin segmentation in ALL and considering these features. It would be very interesting to show specifically how the parameters they consider (TAD domains, RNA pol stalling, conv T) correlate with promoters/ active transcription first and then evaluate if these metrics show significant deviations (enrichment) in the areas most widely affected by genomic rearrangement in ALL and how do these differ between frequent breakpoints and rare breakpoints?

13) Perhaps incorporating metrics that include expression of genes in those regions in B-ALL might strengthen this analysis and provide additional insights into the proposed mechanism. Whilst it may be really difficult to obtain such additional RNA seq data – the authors could consider using gene expression metrics from prior SNP array studies of distinct ALL subtypes.

---

## [Author Response]

*Essential revisions:*

1) The scenario in Figure 6 is potentially interesting but goes far beyond what is supported by the actual data presented in the study. The Discussion section dilutes the main findings by additional speculation. A more detailed discussion of their actual data (instead of what could be gleaned from hypothetical future experiments) would be helpful. There is a lot of speculation on the Results section and in particular around findings that build up to this mechanism and it is not clear how the data/analysis supports these observations.

We thank the reviewer for the suggestions on how to improve the presentation of the obtained results in context of previous work. We have reorganized the text to make a clear distinction between these. In addition, the data related to the mechanisms of how transcriptional features may contribute to genetic instability have been split into two figures (Figure 3 focusing on R-loops and Figure 4 presenting data on DNA access). Instead of a separate model figure, we have included summary panels into these figures. New experimental data and analysis from B-lymphoid and pre-B-ALL cells have been added to support the observations, as further detailed below.

2) The manuscript is of potential interest to a broader audience but mainly written for expert readers in computational biology. To make this work accessible to a wider group of scientists that are potentially interested in the topic, it would seem to be necessary to explain the rationale for the use of certain methods and techniques in some meaningful detail. For instance, the use of t-SNE blots (as for CyTOF) to correlate AID, RAG1 and RAG2 expression with cytogenetic subtypes is of interest, but not discussed what exactly can be seen in the diagrams and why this method was chosen.

We thank the reviewers for acknowledging the interest to a broad audience. We have expanded the rationale for method selection in the Results text and Materials and methods section, as suggested, and explain more in detail what is shown in the diagrams in the respective figure legends, to help the reader follow the analysis performed.

3) Expression of AID and RAG1/RAG2 "markedly distinct between pre-B ALL subtypes": The measurements were performed in fully established ALL clones. The interpretation that different expression levels of AID and RAG1/RAG2 point to a different role of these enzymes in different cytogenetic subtypes is likely incorrect. As in B cell lymphoma, AID and RAG1/RAG2 act together in a multi-step process of clonal evolution towards full transformation. In the fully established leukemia, secondary genetic events and consequences of oncogenic signaling may alter expression levels and obscure the role of AID and RAG1/RAG2 in these leukemia subsets. In addition, the significance of AID and RAG1/RAG2 expression only measured at the mRNA level is unclear. Western blot analyses should be performed if authors feel strongly about documenting different expression levels.

The reviewer raises an interesting point. The clinical relevance of comparing RNA expression in samples from fully established leukemia (i.e. diagnostic samples) between subtypes, has been addressed and supported with patient data in the earlier work by Swaminathan et. al 2015 Nat Immunol. Specifically, in their Results section Swaminathan et al. write “We correlated AICDA mRNA expression in patients at diagnosis with overall and relapse-free survival. […] Notably, higher-than-median AICDA expression was strongly indicative of poor overall patient survival(P = 0.026; Supplementary Figure 1, left [shown using The ECOG E2993 trial that included 215 patients: 106 bone marrow samples and 109 peripheral blood samples]). Likewise, higher-than-median RAG1 mRNA abundance was indicative of shorter relapse-free and overall survival of patients with ALL (Supplementary Figure 1, center and right). In a comparison of AICDA mRNA expression in matched sample pairs at diagnosis and relapse, for most patients, AICDA mRNA abundance was increased at relapse (Supplementary Figure 1 [shown using 49 patients relapsed after successful initial therapy from COG P9906 trial]). These findings suggested that AICDA expression at diagnosis can be used to predict the outcome of patients with ALL.”

As these results convincingly show the relevance of mRNA level of RAG and AICDA in clinical context, we did not include the same analysis to this manuscript. Further confirmation on protein level would be important for further elucidating their functional role at the overt disease step, however such study is beyond the scope of this manuscript.

Here, we provide a first comparison of AID and RAG expression across the cytogenetic subtypes of pre-B-ALL and in context of unsupervised clustering of molecular profiles. Our result shows that specific pre-B-ALL cytogenetic subtypes differ in the prevalence of high RAG or high AICDA mRNA expression that has not been studied before. In particular, the finding that high AICDA expression is prevalent in the subtype designated “other” by the cytogenetic profile is novel. Moreover, we also observed that samples belonging to the high risk subtype often had high AICDA expression, providing further confirmation to conclusions presented in the earlier study regarding clinical relevance.

We have clarified the distinction between the analysis SV sites in context of transcriptional features, and the differences in RAG and AID expression between the subtypes, by placing the expression profile analysis at the end of the manuscript (paragraph starting “To elucidate the potential for RAG and AID mediated genetic instability in leukemia blasts, we compared the expression of the genes RAG1, RAG2 and AICDA across a transcriptome data set”).

4) Do the proposed transcriptional features of instability at SV-sites coincide with cryptic RSS or minimal RAG1/RAG2 substrates? This would add to mechanistic plausibility of their scenario.

The distinction between R-breakp and NR-breakp (no RSS or minimal motif match) was perhaps not explicit from our previous Methods description. We have modified the text with a more detailed explanation on how the RSS status was assigned. Figure 4 in the previous version compared the overlap of the transcriptional features separately for R-breakp and NR-breakp sites in using the ETV6-RUNX1 data on SV (WGS dataset from Papaemmanuil et al. 2014). These data are now presented in Figure 3 and Figure 4 (no RSS vs. RSS/cryptic RSS, respectively, see also Figure 2—figure supplement 3 showing TAD analysis stratified by RSS status), to make the distinction more clear. Further, we extended the analysis by performing the RSS motif search for all breakpoint data used, based on hexameric motifs identified by Papaemmanuil et al. We have included the description in the Methods text and the results were included to Tables [Supplementary-material SD3-data] and [Supplementary-material SD4-data], and [Supplementary-material SD9-data].

5) For the TAD Analysis: Would TAD definition change if a lymphoid origin cell lines or tissue types were used? The authors divide TADS in quartiles on the basis of number of breakpoints within TADs and study how this correlates with% study of convT at these loci. The size range and density of the TADs can vary greatly (T2). The authors should ensure that TAD size does not confound this analysis.

There appears to be a misunderstanding regarding the division of TADs into quartiles. This division was done based on the number of breakp per bp, i.e. considering the TAD size as indicated in the Figure 2 legend “The percentage of TAD spanned by convT in pre-B/B-lymphoid cells is summarized as boxplots from TADs divided into quartiles based on number of breakpoints per bp”. The example TADs shown represent TADs with highest absolute number of breakpoints. We apologize for the possible confusion and have clarified this in the text and figure legend.

To have better consistency across data sources, we repeated the TAD analysis using HiC data from deeply sequenced B-lymphoblastoid cells that recently became available and substituted these results to Figure 1 and Figure 2 (and relevant supplementary results). The results remained essentially the same, typically slight changes in coordinates occurred at TAD boundaries. Compared to the previous, 65% of B-lineage TADs were similar by size using a strict 95% identity cut-off. Figure 6 shows the TAD regions from chr1 (Figure 1) as an example: results from pooled data across cell types is shown above and B-lineage data below (the TAD with several breakpoints is the 3rd from left).

Author response image 1.**DOI:**
http://dx.doi.org/10.7554/eLife.13087.036

6) Correlation of 'transcriptional features' such as prevalence of POL2 stalling, DNA sequences that are susceptible to R-LOOP formation and also convergent transcription, with regions that are frequently rearranged in ALL, and in regions with high prevalence of RSS.

The referee suggests a valid approach that is very similar to what we have carried out based on enrichment metrics. As the data is essentially discrete (presence/absence of feature or breakpoint counts), we carried out statistical analysis using binomial and hypergeometric tests. In parallel, empirical estimation of random overlap frequency served as an additional confirmation.

Analysis that stratifies breakpoint sites based on the recurrence (frequent rearrangements in ALL) was presented in Figure 4 for the ETV6-RUNX1 subtype (now shown in Figure 4). Specifically, breakpoints were binned into hotspots using a 1 kb distance. We added new data to confirm that similar results were obtained by considering all WGS datasets included to our study (included to [Supplementary-material SD3-data] and [Supplementary-material SD9-data]). This analysis reveals whether higher overlap with transcriptional features can be found at highly recurrent SV sites. Following the suggestion from the reviewer, we extended the analysis of breakpoint hotspots with analysis of R-loop forming sequences (refer to Figure 3 and Figure 4). The RLFS sequence motif alone did not display a strong enrichment trend for recurrent sites. However, we found that convT can contribute to R-loop generation based on DRIP-seq data (Figure 3), and that convT overlap was increased at recurrent breakpoints. Although the experimental evidence for R-loops used in our analysis comes from non-B-lineage cells (ES cells), the biochemical mechanism for recruitment of AID or similar complexes binding to R-loops is unlikely to be cell-specific. Overall, our results are consistent with the mechanism proposed for convT/Pol2 stalling mediated AID recruitment in lymphomas (Pavri et al. 2010, Meng et al. 2014, Wang et al. 2014), and we have included experimental and sequence-level motif data to support a mechanistic link between genetic instability and convT and Pol2 stalling. The new Figure 4 and Figure 3 address the difference between overlap of SV sites carrying RSS-motifs and those lacking them, respectively.

7) Whilst there is a notable enrichment against other active promoters for example the 'width' of the 'stalling' region, although what this really represents, as defined from GRO-seq data, is the high density of active RNA polymerases, so is it possible that these represent regions of active transcription.

We thank the referee for raising this important question. A transcript that would have uniform signal in GRO-seq from its TSS to TTS would correspond to a genomic region where the Pol2 elongation rate is uniform. However, this is rarely observed (reviewed in Jonkers and Lis, Nature Reviews Molecular Cell Biology, 16, 167–177, 2015). A Pol2 stalling region corresponds to elevated signal within an actively transcribed region. As described in the review by Jonkers and Lis, Pol2 has been documented to slow down e.g. at exons (to facilitate splicing), R-loops (difficult structurally to transcribe) and is actively paused at TSS regions (regulated by phosphorylation, this type of pausing typically occurs within the first 50-200 bp). Each of these events are visible as a signal increase in GRO-seq. Our changepoint analysis assigns a relative increase in the signal (cut-off at highest 90%) within a transcribed region as a Pol2 stalling site, which corresponds to local slowing of Pol2. Notice that the signal data is scaled and analyzed separately for different gene regions.

One important control that we are observing real stalling events is presented in the new Figure 3 (previously in Figure 5): the detected Pol2 stalling events overlap RLFS motifs more often than expected by random. RLFS contribute to the stalling event itself due to slow progression at R-loops. We also show data (now in Figure 3—figure supplement 1) that the presence of RLFS is visible as statistically significant elevation of GRO-seq signal. As additional confirmation, these results were reproduced using B-lineage and ES data (see Figure 3 and Figure 3—figure supplement 1).

As additional experimental data that could assist in interpreting the GRO-seq signal profile, we performed Pol2 ChIP-seq using antibodies against the Ser2 and Ser5 phosphorylated forms. Ser5 phosphorylation is present before the Pol2 is released to active elongation (Zhou et al. 2012). Using a similar analysis of the signal profile to detect Pol2 stalling from the ChIP-seq profile, we found highest overlap of RSS-motif containing SV sites with the Ser5 form, typically considered as the paused/stalled Pol2. However, also locally elevated Ser2 phosphorylated Pol2 has been found to be relevant for genomic instability (Hatchi et al. 2015): in this case the stalling typically occurs at gene ends. We have included new references to literature in the Introduction, Methods and Discussion to assist in interpretation of the signal profiles presented.

The width of Pol2 stalling was analyzed since displaced nucleosomes in vicinity of Pol2 could enhance access to RSS sites and thereby RAG cleavage events. To support this conclusions, we performed additional analysis (see new Figure 4) to characterize more in detail the stalling regions. DNA accessibility, based on DNAse-seq peaks, was increased at Pol2 stalling sites (new Figure 4). Specifically, we compared peaks from different gene regions, by dividing them based on overlap with Pol2 stalling. Consistently, the peak width and intensity was higher for the peaks overlapping Pol2 stalling. We carried out the analysis with both B-lineage and ES cell data, to support our rationale that the general properties of the Pol2 stalling sites are cell-type-independent (see also comment (9)).

As an additional control, we correlated the width of Pol2 stalling at the TSS region and the expression level (rpkm within the gene body). There was weak negative correlation, indicating that the vulnerability of TSS with wide stalling is not simply due to higher expression level.

8) With previous reports showing enrichment of genomic rearrangements at active promoters and enhancers it is not clear whether the observations here are a consequence of transcriptional activity in these regions or an enriched mechanism underpinning the regions that are frequently targeted. The authors should present a global analysis – see statistical review section.

It appears that a similar concern is raised here as in (7): whether high transcription activity alone explains the overlap. We addressed this question in two ways:

In the TAD analysis, we performed an additional control where we used the ENCODE chromatin segmentation for defining active promoters and enhancers. Subsequently, we overlapped these regions with convT/Pol2 stalling to obtain a measure for the base pair span i) with convT/Pol2 stalling overlap and ii) without overlap. Next, the% of TADs spanned by each feature type was compared between breakpoint frequency quartiles. There was a consistent increasing trend (see new Figure 2—figure supplement 6) for the convT/Pol2 stalling overlapping parts of active promoter and enhancer regions, thus pinpointing the most vulnerable regions.

We analyzed breakpoint hotspot regions that localize to annotated gene regions. The genes were binned by the transcriptional level to control for the effect of transcriptional activity. Subsequently, the significance of overlap with the transcriptional features was evaluated for the different bins. The result was consistent across a wide range of expression levels and this new data has been included to [Supplementary-material SD9-data].

The new analysis results support the conclusion that convT/Pol2 stalling are enriched at regions that are frequently targeted.

9) A major limitation of the analysis here is that there are not overlapping datasets from the same samples. As the authors have access to direct primary patient cells and cell lines representative of B-ALL their interpretations of the data would significantly benefit from performing RNAseq, POL2 chip, and potentially MNase-seq to support their observations (Pol2 stalling and convergent transcription).

We have carefully considered this comment and added new experimental validation and supporting data from pre-B-ALL and B-lineage cells. However, we would like to stress that the Pol2 transcription process itself should not fundamentally differ between leukemic and non-leukemic cells. Characteristic and reproducible transcriptional features at specific genomic regions can be identified by comparing samples of different origin. Here, we have identified where Pol2 stalling and convT occurs from pre-B-ALL patient and cell line material that represent several different subtypes, from normal B-lymphoid cells and ES cells. We then related the features detected from B-lineage cells to SV event frequency in pre-B-ALL patients on a genome-wide level (Figure 2–Figure 4), while utilizing the ES and B-lineage data jointly to discern what properties of convT and Pol2 stalling regions may leave the DNA vulnerable to SV.

Following the reviewer’s suggestion, we performed additional Pol2 ChIP-seq data to support the detection of Pol2 stalling events. We also considered MNase-seq but obtaining high enough sequence depth was deemed unfeasible. As an alternative, we tested a ChIP protocol that involves MNase digestion. The signal profile obtained therefore reflects Pol2 enrichment on the background of nucleosome positions. We focused here on the overall signal profile across gene regions (Pol2 stalling detection based on change points) but we make the data available for further analysis that also models the nucleosome positioning. RNA-seq was not included as the signal does not reflect active ongoing transcription (mainly mature RNAs are measured) and is additionally influenced by RNA stability, which would complicate the analysis.

We have also extended the results evaluating the properties of regions with Pol2 stalling with more data from pre-B-ALL and B-lineage cells (included to Figure 3 and Figure 4). To address DNA access, we utilized the ENCODE DNAse-seq profiles. Acquiring complementary data from primary ALL cells would represent a significant investment and according to our experience, primary patient material is not available at sufficient amounts to allow performing each assay.

10) Overall there is a clear narrative issue, the dataset is complex and the analysis is not clear nor comprehensive – making it rather hard for the reviewer and in time a reader to comprehensively evaluate the analytical approaches used, as well as, the evaluate the interpretation of the findings, which often render conclusions as facts by citing other papers rather than being supported by the data in itself. It would be very helpful if the authors included a supplemental figure containing a flow chart describing what datasets where put together and which subsets of data were used for which analysis.

We thank the reviewer for the suggestions how to improve the manuscript. We have reorganized the manuscript and modified Figure 1 to highlight the main datasets and added a new supplementary figure to illustrate the data integration approach more in detail (see Figure 1—figure supplement 2). To clarify the interpretations, we have made a more clear separation between the Results and Discussion sections.

11) Results should be interpreted on the basis of their analysis. Conclusions that account or include findings from the literature should be placed in the Discussion section.

We have re-organized the Results and Discussion text to address this.

12) It would be useful if the authors performed a global analysis- accounting for chromatin segmentation in ALL and considering these features. It would be very interesting to show specifically how the parameters they consider (TAD domains, RNA pol stalling, conv T) correlate with promoters/ active transcription first and then evaluate if these metrics show significant deviations (enrichment) in the areas most widely affected by genomic rearrangement in ALL and how do these differ between frequent breakpoints and rare breakpoints?

We thank the reviewer for the suggested use of chromatin segmentation data and inclusion of additional results to further support the relevance of transcriptional activity. We added a new figure (Figure 1—figure supplement 3) that compares the expression median level of genes within TADs (using the pre-B-ALL transcriptome data) between TADs binned by breakpoint frequency.

Next, we utilized the chromatin segmentation generated by the ENCODE consortium that is based on a large collection of ChIP-seq data from B-lymphoblastoid cells (unfortunately a similar segmentation is not available directly from ALL cells). Similar as in Figure 2, we compared their span within TADs sorted by breakpoint frequency (included in [Supplementary-material SD2-data]). These analyses confirmed that TADs with high number of breakpoints have more overlap with chromatin segments representing active transcription (including active promoters), supporting a transcription-coupled mechanism for the observed genetic instability.

Subsequently, we analyzed whether the convT/Pol2 stalling had relevance as a specific feature within active promoters and enhancers defined by the chromatin segmentation. These results are presented as an additional figure supplement to Figure 2 (see new Figure 2—figure supplement 6). The presence of convT/Pol2 stalling was an important determinant for the promoter and enhancer enrichment within TADs that contain most breakpoints.

We deemed these two approaches appropriate to address first the relevance of transcription and then to distinguish the specific contribution of convT/Pol2 stalling. Correlating convT/Pol2 stalling with active transcription is not meaningful as, by definition, these transcriptional features occur within areas of active transcription. We correlated the width of Pol2 stalling at the TSS region with the expression level of the transcript. Comparison of frequent and rare breakpoints is presented in Figure 3, Figure 4 and [Supplementary-material SD3-data] and [Supplementary-material SD9-data].

*13) Perhaps incorporating metrics that include expression of genes in those regions in B-ALL might strengthen this analysis and provide additional insights into the proposed mechanism. Whilst it may be really difficult to obtain such additional RNA seq data – the authors could consider using gene expression metrics from prior SNP array studies of distinct ALL subtypes.*

Adding expression metrics from pre-B-ALL is an excellent suggestion. By “those regions” we understand the reviewer refers to TADs and chromatin segmentation data. We analyzed expression within TADs by calculating the median and maximum expression level in pre-B-ALL cells for the co-localized genes. To this end, we utilized the 1382 patient transcriptomes available in our integrated dataset across pre-B-ALL studies (these expression metrics are included in [Supplementary-material SD1-data]). As a second measure, we used the total GRO-seq read count per TAD. A new figure has been added to show how the activity of transcription increases in the TADs with highest breakpoint frequency (Figure 1—figure supplement 3), supporting a transcription-coupled mechanism. We have further provided a new control that shows the% of transcribed region spanned by convT/Pol2 stalling (new Figure 2—figure supplement 4). This result confirmed that the enrichments we see are not just a reflection of total transcriptional activity. We also binned genes by their transcription level for the analysis of recurrent breakpoints, similarly to control for the effect of transcription (new [Supplementary-material SD9-data]).

The new analysis results support the conclusion that convT/Pol2 stalling are enriched properties of regions that are frequently targeted.